# Bioinspired bio-voltage memristors

Tianda Fu [1], Xiaomeng Liu [1], Hongyan Gao[1], Joy E. Ward[2], Xiaorong Liu[3], Bing Yin[1], Zhongrui Wang[1], Ye Zhuo [1], David J. F. Walker[2], J. Joshua Yang [1], Jianhan Chen[3,4,5], Derek R. Lovley[2,4] & Jun Yao[1,4✉]

Memristive devices are promising candidates to emulate biological computing. However, the typical switching voltages (0.2-2 V) in previously described devices are much higher than the amplitude in biological counterparts. Here we demonstrate a type of diffusive memristor, fabricated from the protein nanowires harvested from the bacterium *Geobacter sulfurreducens*, that functions at the biological voltages of 40-100 mV. Memristive function at biological voltages is possible because the protein nanowires catalyze metallization. Artificial neurons built from these memristors not only function at biological action potentials (e.g., 100 mV, 1 ms) but also exhibit temporal integration close to that in biological neurons. The potential of using the memristor to directly process biosensing signals is also demonstrated.

[1] Department of Electrical and Computer Engineering, University of Massachusetts, Amherst, MA 01003, USA. [2] Department of Microbiology, University of Massachusetts, Amherst, MA 01003, USA. [3] Department of Chemistry, University of Massachusetts, Amherst, MA 01003, USA. [4] Institute for Applied Life Sciences (IALS), University of Massachusetts, Amherst, MA 01003, USA. [5] Department of Biochemistry and Molecular Biology, University of Massachusetts, Amherst, MA 01003, USA. ✉email: juny@umass.edu

Biological brains are highly efficient in signal processing and intelligent decision making. Functional emulation can thus lead to advanced computing systems[1,2]. The ability to register accumulative stimuli (e.g., charge flux) by a state variable (e.g., conductance) enables typical memristors to mimic neuromorphic behaviors that are inherently history-dependent[3–8]. Various synaptic devices, neuromorphic components, and computing architectures have been demonstrated based on this property[1,2,9–11]. Recently, a type of diffusive memristors has been shown to complement nonvolatile or drift memristors[3]. The process of conductance decay in the diffusive memristors effectively produces an 'internal clock' to encode the relative temporal information, leading to functional emulations in short-memory synapses[3,12,13], artificial neurons[14], and capacitive neural networks[15]. However, the switching voltages (0.2–2 V) required for all previously described memristors[3–8] are higher than the amplitude of 50–120 mV in biological counterparts[16]. Power consumption typically scales quadratically with signal amplitude, and thus functional emulation with memristors has yet to attain the extraordinary low power requirements of biosystems[11]. A field-driven mechanism in many filamentary memristors[8,17] means that device scaling may not necessarily reduce the switching voltage, and hence the power demands for high-density applications. On the other hand, the functional similarity of memristors to biological systems may facilitate electronic-biological interfaces[18–20]. However, the functional similarity without parameter matching still requires additional circuitry for interfacing, adding costs to the vision of a seamless integration[21–23]. Therefore, neuromorphic devices functioning at biological voltages (bio-voltage, e.g., ≤100 mV) are expected to not only lower the power requirements, but also enable communicative electronic-biological interfaces.

## Results

**General concept**. Memristive switching is often an electrochemical process associated with an ionic/valence state change in the dielectric layer[1–8]. For example, a three-step process of anodic oxidation ($M \rightarrow M^+ + e$), $M^+$ migration, and cathodic reduction ($M^+ + e \rightarrow M$) is typically involved in the switching dynamics in metallization memristors[8]. The active metals (M) involved are generally readily oxidized to ions in the ambient environment[24], and ion migration is usually not a threshold event. Therefore, the cathodic reduction is expected to be the major factor affecting the switching voltage. Thus, a catalyst that can lower the reduction overpotential[25] (Fig. 1a) should reduce the switching voltage in memristors (Fig. 1b, c).

The protein nanowires of the bacterium G. sulfurreducens (Fig. 1d) are well known for their ability to facilitate metal reduction[26–28] and they may facilitate $Ag^+$ reduction[29]. As the reduction takes place in a biological environment, it indicates that the protein nanowires may be able to catalyze bio-voltage memristors (Fig. 1e). Therefore, we constructed Ag memristors with protein nanowires harvested from G. sulfurreducens to determine if they would facilitate the memristive switching.

**Device structures and characterizations**. Two memristor device configurations fabricated with protein nanowires were studied. The first one was comprised of a pair of Ag electrodes (~200 nm spacing) on an insulating (SiO₂/Si) substrate. A thin (~500 nm)

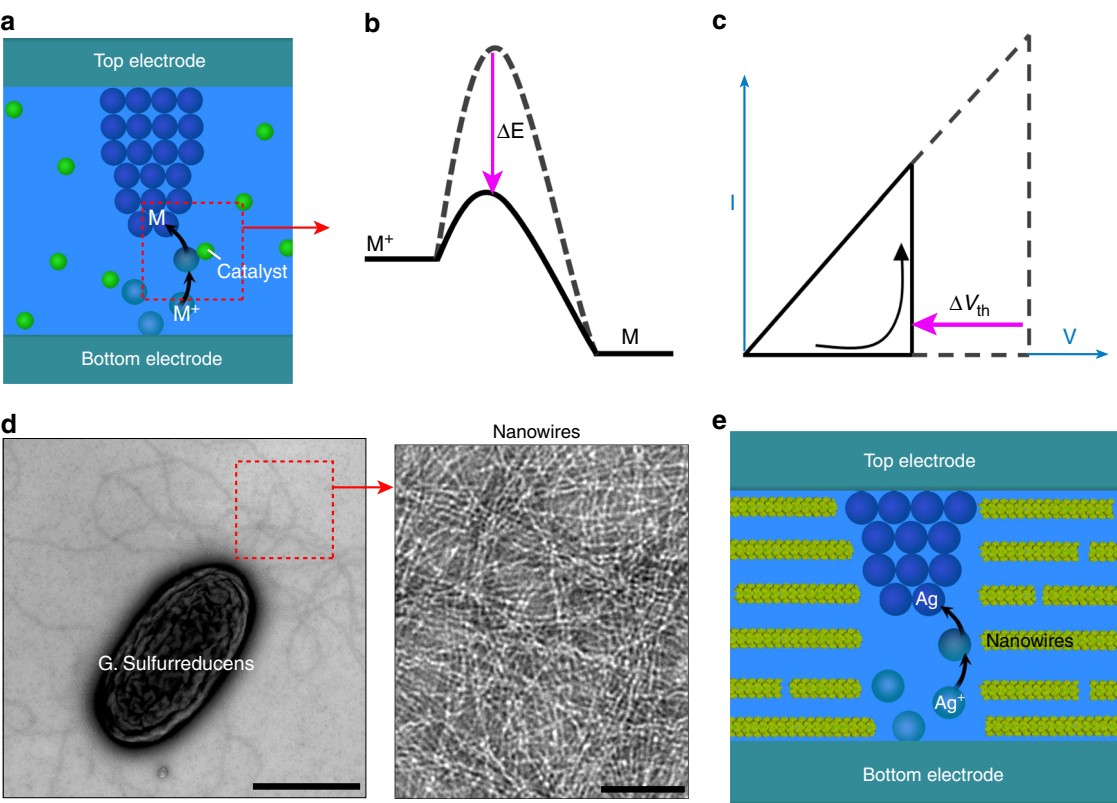

**Fig. 1 Proposal of catalyzing bio-voltage memristors. a** Schematic of an introduced catalyst (green dot) in a memristor that facilitates the cathodic reduction by (**b**) bringing down the reduction overpotential (ΔE), which leads to (**c**) a decrease in the switching voltage (ΔV$_{th}$). **d** TEM images of a G. sulfurreducens and purified protein nanowires (right) harvested from G. sulfurreducens. Scale bars, 1 μm (left) and 100 nm (right). **e** Schematic of introduced protein nanowires in a memristor that facilitate the cathodic reduction of $Ag^+$ to attain possible bio-voltage switching.

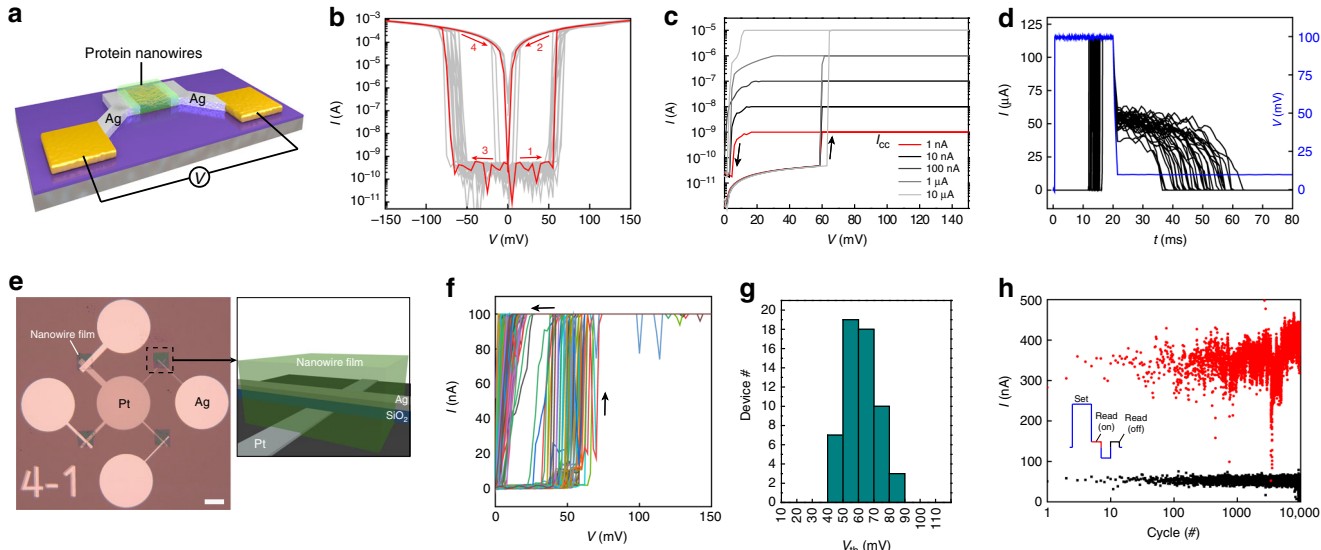

**Fig. 2 Protein-nanowire devices and memristive switching. a** Schematic of the planar device structure. **b** Representative *I–V* curves from a device. The red curves show a complete sweep loop from 0→150 mV →0 →-150 mV →0. **c** Switching *I–V* curves from a device with the current compliance ($I_{CC}$) set from 10 μA to 1 nA. **d** Pulsed measurements in a device (N = 32). Same pulses (100 mV, 20 ms; blue curve) were applied to the device with the current (black curves) measured. The conduction state in the device was continuously monitored after the pulse by using a 10 mV reading voltage. **e** Photo image of 4 vertical protein-nanowire devices with varying sizes of 20 × 20, 10 × 10, 5 × 5, and 2 × 2 μm². Scale bar 50 μm. The schematic shows the layered Ag–SiO₂–Pt (100–25–30 nm thick) device structure embedded in patterned protein-nanowire film (~500 nm thick). **f** A continuous 800 *I–V* sweeps in a vertical protein-nanowire device. Above measurements were done in the ambient environment with RH~35%. **g** Statistics of the turn-on voltage from 57 vertical devices, with an average value of 58 ± 10 mV (± s.d.). **h** A 10⁴ cycles from a vertical device. Each cycle involved a pulse wave form (inset) consisting of a programming (set) pulse (100 mV, 40 ms), a read pulse (10 mV, 19 ms) of the On current, a reset pulse (−50 mV, 20 ms) to facilitate the relaxation, and a read pulse (10 mV, 19 ms) of the Off current. Current compliance was applied to the On read current with a load resistor. The Off read current was at the instrument noise level (<100 nA) for the measurement range used.

layer of protein nanowires was deposited in the nanogap (Fig. 2a; Supplementary Fig. 1 and Methods). Scanning electron microscope (SEM) and transmission electron microscope (TEM) images revealed that the nanowires were densely packed in the film (Supplementary Fig. 1c, d). Molecular dynamics simulations (Supplementary Fig. 2) indicated that the inter-wire spacing is at sub-nanometer scale[30], which is similar to that of the grain boundary in inorganic dielectrics. As a result, the nanowire film behaved like a dielectric (Supplementary Fig. 1c) and was demonstrated to be compatible with lithographic patterning (Supplementary Fig. 3). The conductivity of the protein nanowires depends on the pH at which they are prepared[31]. We used protein nanowires of relatively low conductivity (Supplementary Fig. 4), which acted like a dielectric employed in electrochemical metallization memristors[8]. After electroforming (Supplementary Fig. 5), the device formed stable unipolar switching in the ambient environment (Fig. 2b). In a representative sweep loop (red curves), the device transitioned to a low-resistance state (LRS) at 60 ± 4 mV (±s.d.), which spontaneously relaxed to a high-resistance state (HRS) at close-to-zero bias. The following sweep in the negative bias showed symmetric behavior, with the turn-on voltage at −65 ± 5 mV (±s.d.). Statistics over 60 planar devices having different electrode spacing (e.g., 100–500 nm, each set 12 devices) revealed a consistent distribution of average turn-on voltage between 45–80 mV (Supplementary Fig. 6). This is at least 3-fold lower than the lowest values in previously described diffusive memristors[3,5,13], and is in the range of biological action potentials[16]. The large On-Off ratio (e.g., ~10⁶) in the device, for a reduced switching window, led to a sharp turn-on slope ~0.4 mV/dec (Supplementary Fig. 7), which is the sharpest demonstrated to date[3,5]. The switching voltage was largely independent of programming current, which maintained at ~60 mV with current compliance ($I_{cc}$) reduced to 1 nA (Fig. 2c), indicating a

field-dominant mechanism. The $I_{cc}$ was among the lowest in memristors[5], which, due to further reduced programming voltage, indicates the potential for low-power operation. The volatile switching was further elucidated with pulse measurements (Fig. 2d), during which fixed voltage pulses (100 mV) yielded stochastic incubation delay ($\tau_d = 13 ± 1$ ms, ±s.d.) and relaxation ($\tau_r = 29 ± 7$ ms, ±s.d.). $\tau_d$ and $\tau_r$ are close to time scale in biological responses such as synaptic Ca²⁺ rise and decay[32]; longer pulses yielded longer relaxation (Supplementary Fig. 8), indicating the evolving dynamics in conduction that is desirable for biological emulations.

Vertical device configuration, which is considered preferable for device integration, was also examined. The top (Ag) and bottom (Pt) electrodes were separated by a 25 nm-thick SiO₂ layer, which was further embedded in the protein-nanowire film (Fig. 2e; Methods). The vertical device is topologically the same as a planar one, as the exposed SiO₂ vertical edge defines a vertical insulating substrate. An edged device structure was employed previously in both device characterization and crossbar integration[33,34]. As the electroforming is typically field-driven, a closer electrode spacing in the vertical device resulted in almost forming-free switching (Supplementary Fig. 9a). The average forming voltage ~70 ± 19 mV (±s.d.) (Supplementary Fig. 9b) was even smaller than the switching voltages in previous memristors[3,5,13]. The formed device showed a narrow distribution of turn-on voltages at ~55 mV (Fig. 2f). Statistics of 60 vertical devices showed a high yield of ~95% (57/60). The turn-on voltages were 40–80 mV (Fig. 2g), which is consistent with results from planar devices (Supplementary Fig. 6). The device was repeatedly programmed (with 100-mV pulses) for at least 10⁴ cycles (Fig. 2h), showing robust endurance over typical organic or soft-material memristors[35]. The vertical structure also enabled investigations into device scaling. The devices showed similar

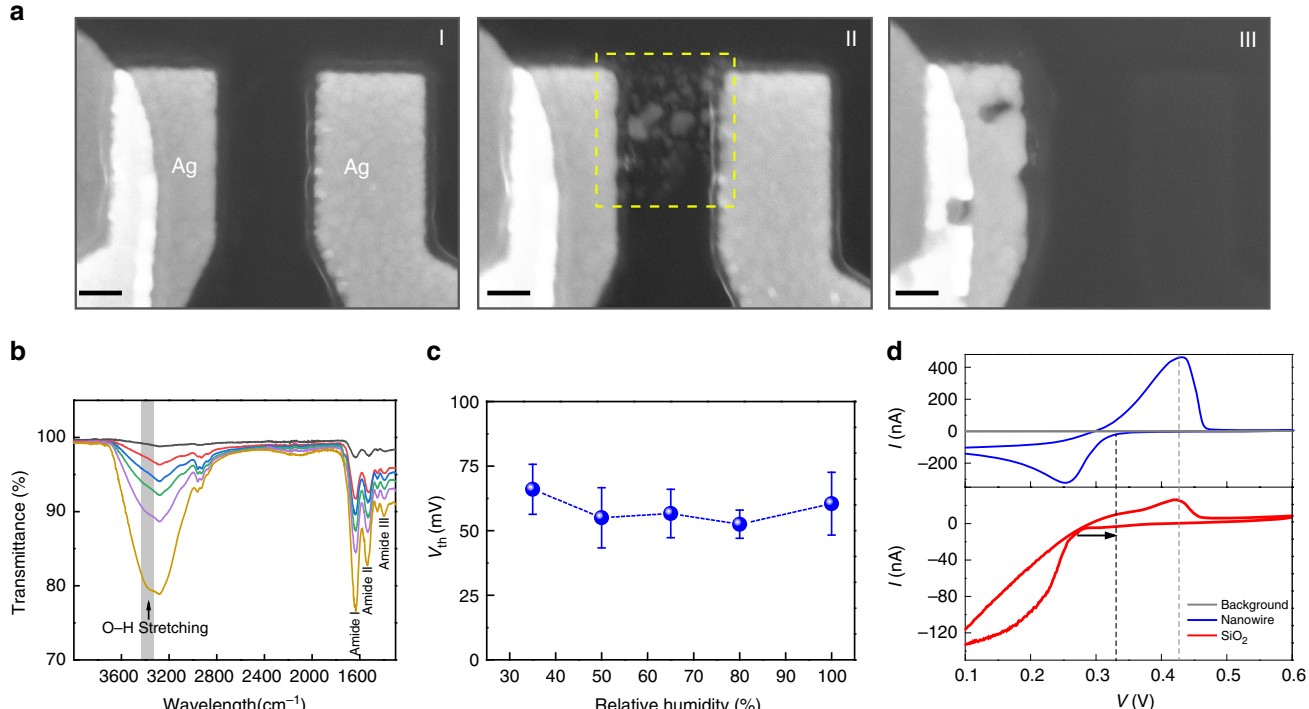

**Fig. 3 Conducting filament and switching dynamics. a** SEM images of a protein-nanowire device (I) before, (II) after electroforming, and (III) after removal of the protein nanowires with ultrasonication. Scale bars, 100 nm. **b** FTIR spectra of a protein-nanowire film (~200 nm thick) at relative humidity (RH) of 0% (gray), 35% (red), 50% (blue), 65% (green), 80% (purple) and 100% (yellow). The broad peaks ~3400 $cm^{-1}$ correspond to O-H stretching band in free water[42]. The increased intensity indicates increased water adsorption in the film at higher RH. The increased intensities in other amide peaks could be caused by protein segments that became more mobile after moisture filling interstitial voids. **c** Average turn-on voltage ($V_{th}$) from devices ($N = 12$) in different RH. The error bars represent the standard deviation (s.d.). **d** Cyclic voltammetry curves using protein-nanowire coated Au cathode (blue) and $SiO_2$-coated Au cathode (red) in $Ag^+$ (5 mM) solution at a rate of 25 mV·s$^{-1}$. The background curve (gray) was performed without $Ag^+$. The onset reduction potential shows a right shift (black arrow) from ~0.25 V to ~0.35 V when switching from $SiO_2$ coating to protein-nanowire coating in the Au cathode, whereas the oxidation peak (~0.43 V) does not shift.

switching behaviors when the size was reduced from $20 \times 20\ \mu m^2$ to $2 \times 2\ \mu m^2$ (Supplementary Fig. 10), indicating a localized filamentary switching nature. Similar switching was obtained in devices with sizes as small as $\sim 100 \times 100\ nm^2$ (Supplementary Fig. 11), demonstrating the potential in device scaling for high-density integration.

**Mechanistic investigations.** Two types of protein nanowires can be recovered from G. sulfurreducens, protein wires that assemble from the pilin monomer PilA and wires that assemble from the c-type cytochrome OmcS[28]. The relative abundance of each type of wires depends upon the conditions under which the cells are grown. The protein nanowires in our preparations (Supplementary Fig. 1e) had an average diameter of $2.9 \pm 0.35\ nm$ ($\pm$s.d.), which is consistent with the 3 nm diameter of pilin-based nanowires[28] and inconsistent with the 4 nm diameter of protein nanowires comprised of OmcS[36,37]. Furthermore, devices constructed with protein nanowires harvested from a strain of G. sulfurreducens in which the gene for OmcS was deleted yielded similar results (Supplementary Fig. 12). These results suggested that pilin-based protein nanowires were the important functional components.

The protein nanowires, as hypothesized, were found to play the central role in realizing the bio-voltage switching. First, the possible contribution from chemical residue in the protein-nanowire solution was excluded (Supplementary Fig. 13). Second, the protein nanowires helped to reduce the electroforming voltage substantially (e.g., <2 V; Supplementary Fig. 14a), compared to

that of a pair of bare Ag electrodes on $SiO_2$ (e.g., >15 V; Supplementary Fig. 14b) in planar devices. Similarly, vertical devices without protein nanowires could not yield low-voltage forming or switching (Supplementary Fig. 15). Although surface defects introduced by material deposition can facilitate electroforming[33], such a large reduction indicates that the conduction channel may be built up entirely in the protein nanowire film. We used a device with a larger electrode spacing (~500 nm) to facilitate observation of the conduction channel (Fig. 3a, I). After electroforming, Ag nanoparticles were distributed between the electrodes (Fig. 3a, II). This observation is consistent with the general proposed mechanism for Ag-based memristors in which Ag nanoparticles are the signature of anodic $Ag^+$ migration and metallization to form the conductive filaments[3,38]. In particular, the spherical configuration of the particles helps to lower the interfacial energy, which is considered to be responsible for the spontaneous filament rupture in a diffusive memristor[3]. When the protein nanowire film was removed with ultrasonication in water, a clean $SiO_2$ substrate without Ag nanoparticles was observed (Fig. 3a, III). These results confirmed that the conduction channel or Ag filament was built up entirely in the protein nanowire film. Control studies indicated that neither the $SiO_2$ surface property (Supplementary Fig. 16) nor the percolation material structure (Supplementary Fig. 17) contributed to the bio-voltage switching.

Additional studies elucidated the catalytic role of protein nanowires. The influence of moisture is important in various inorganic memristors because it affects oxidization rate and cation mobility[39], and hence the switching dynamics[40,41]. The

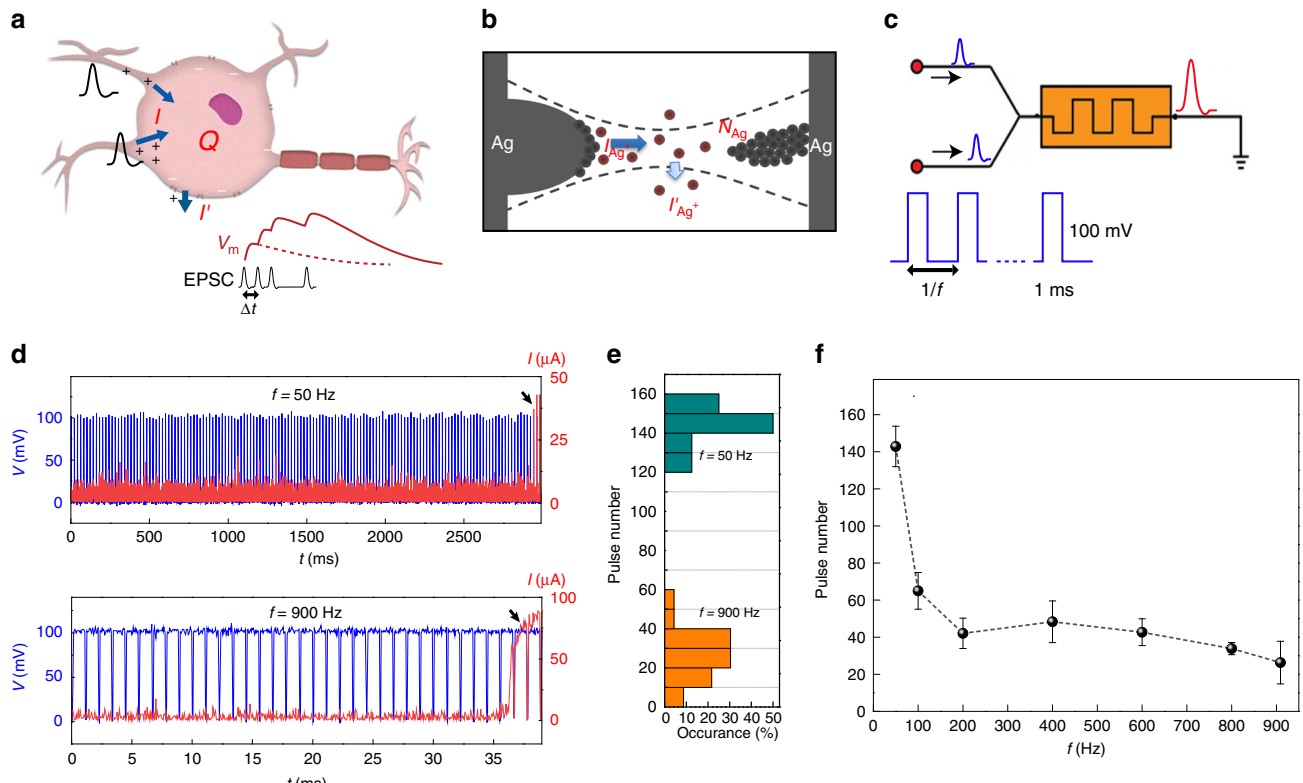

**Fig. 4 Protein nanowire artificial neuron displaying temporal integration of inputs. a** Schematic of a biological neuron integrating excitatory postsynaptic current (EPSC). $I$ denotes the injected EPSC, $I'$ denotes the leaky current through the cell membrane, and $Q$ denotes the net cytosolic charge. The bottom panel is a schematic of temporal integration of EPSC. The membrane potential ($V_m$) shows a close-to-linear summation of short-interval EPSC spikes, whereas the summation deviates for long-interval EPSC spikes. **b** Schematic of proposed dynamics of filament formation in the protein nanowire memristor. The dashed lines delineate the given filamentary volume. $I_{Ag+}$ denotes injected cationic $Ag^+$ current, $I'_{Ag+}$ denotes the $Ag^+$ diffusion (leaky) current out of the filamentary volume, and $N_{Ag}$ denotes the net Ag element in the filamentary volume. **c** Schematic of a leaky integrate-and-fire artificial neuron built from a protein nanowire memristor. Excitatory postsynaptic potential (EPSP) signals are represented by action-potential alike spikes (100 mV, 1 ms) and their temporal correlation is modulated by frequent $f$. **d** The neural firing (turn-on, indicated by black arrows) in a protein nanowire device at low-frequency (50 Hz, top panel) and high-frequency (900 Hz, bottom panel) EPSP signals. **e** Statistics of pulse number required for neural firing for low-frequency (50 Hz, cyan) and high-frequency (900 Hz, orange) EPSP signals. **f** Average pulse number of EPSP signals required for neural firing at different frequencies (50–900 Hz). The error bars represent the standard deviation (s.d.). The pulse number shows a relatively flat trend at high frequency (200–900 Hz), indicating a close-to-linear temporal integration less sensitive to spike interval. The trend deviates at low frequency (≤100 Hz).

protein-nanowire film also adsorbed moisture in the ambient environment[30] (Supplementary Fig. 18). Fourier-transform infrared spectroscopy (FTIR) spectra[42] revealed that moisture adsorption increased with increased relative humidity (Fig. 3b). However, the devices maintained a similar turn-on voltage ($V_{th}$) over a wide range of relative humidity (Fig. 3c; Supplementary Fig. 19), indicating that anodic oxidization and ion migration were not the determining factors in bio-voltage switching. Further evidence that oxidization and ion migration were not the determining component was the finding that turn-on voltage <0.5 V could not be achieved with devices made from polyvinylpyrrolidone (PVP) which adsorbs similar amounts of moisture[43] as protein nanowires (Supplementary Fig. 20). This result is consistent with previous studies in which Ag memristors based on biomaterials or organic materials that could easily adsorb moisture also did not yield switching voltages below 0.5 V[35]. Collectively, these results suggested that the protein nanowires played an important role in the cathodic reduction step rather than anodic oxidation and cation migration.

We therefore carried out studies in decoupled $Ag^+$ reduction in an electrochemical cell using cyclic voltammetry. Previous study revealed a direct electron donation from the dielectric to $Ag^+$ during the reduction or filament formation in Ag memristors[38]. To mimic the dielectric environment of $Ag^+$ reduction in a memristor, the working electrode was coated with a thin layer of dielectric. The cyclic voltammetry measurements showed that, compared to a $SiO_2$ dielectric coating, the protein-nanowire coating yielded a right shift in the reduction peak (Fig. 3d). This right shift or decrease in reduction potential, with respect to an unchanged position in the oxidation peak, demonstrated that protein nanowires facilitate cathodic reduction of $Ag^+$. Protein nanowires have specifically evolved for electron transfer to metals[26–28], but the specific molecular mechanism for $Ag^+$ reduction warrants further investigations.

The bio-voltage memristor studied here falls into the category of electrochemical metallization cells[8], in which the conductance modulation requires a physical morphology change in the metallic filament. A reading voltage below the threshold switching voltage is hardly expected to perturb the conductance, because the physical evolution in the filament requires an electrochemical reduction that is largely a threshold event by overcoming the electrochemical reduction potential as discussed above. These considerations suggest that the bio-voltage memristors have functional stability.

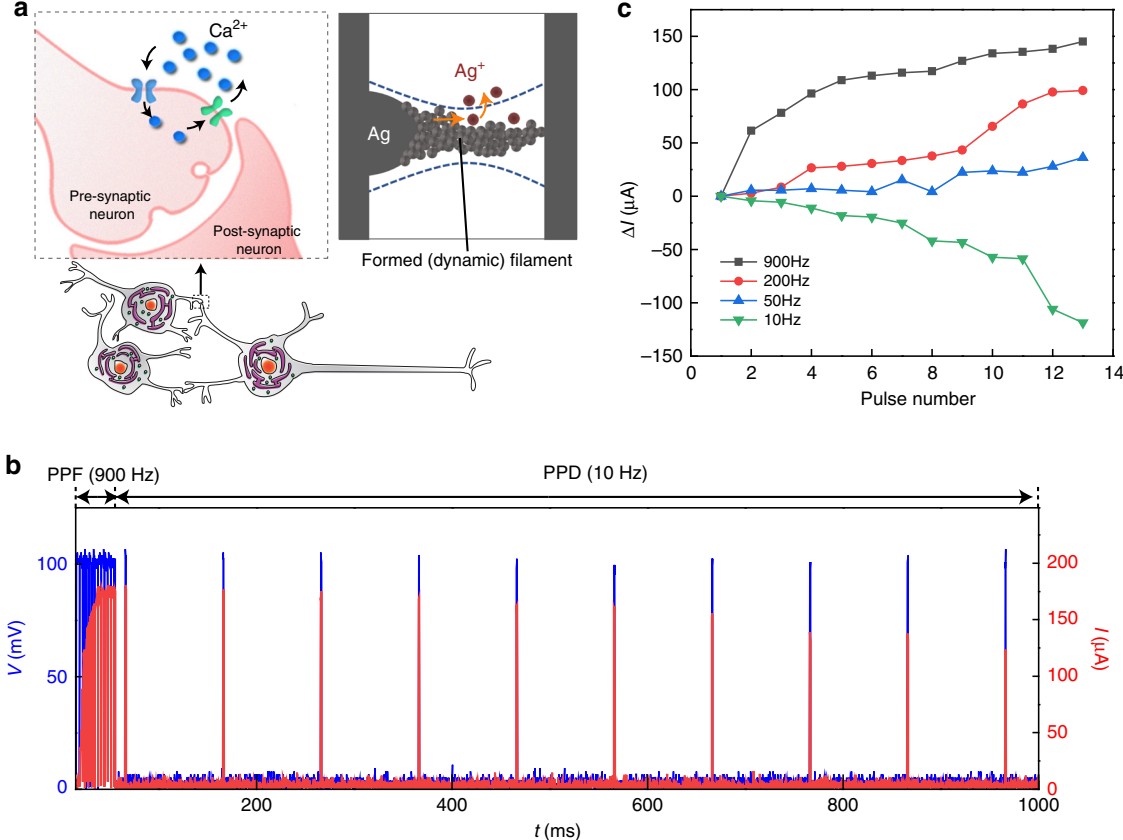

**Fig. 5 Protein nanowire artificial synapse. a** (Left schematic) $Ca^{2+}$ influx and extrusion in a biosynapse that underlies synaptic plasticity. (Right schematic) The increase (by anodic injection) or decrease (by diffusion) of Ag atoms in a formed filament can lead to the dynamic increase or decrease in filament size. The resultant conductance change can mimic the synaptic plasticity. **b** Conductance modulation in an activated protein nanowire memristor by modulating the frequency in inputs (100 mV, 1 ms). Paired-pulse facilitation (PPF) featuring continuous conductance increase was observed at high frequency (900 Hz), whereas paired-pulse depression (PPD) featuring continuous conductance decrease was observed at low frequency (10 Hz). **c** Conductance modulation in a protein nanowire memristor with different input frequencies.

**Bio-voltage artificial neuron.** The protein-nanowire memristors enable the construction of bio-voltage neuromorphic components such as an artificial neuron. The biological neural firing is triggered by membrane potential ($V_m$), which is directly related to net charge ($Q$) in a given cytosolic volume ($Q = C_m \cdot V_m$; $C_m$ is the membrane capacitance); therefore the state dynamics are often modeled[44] by $C_m \frac{dV_m}{dt} = I - g_m V_m$ (Eq. 1), where $I$ denotes injection current and $g_m V_m$ the leaky current related to the membrane conductance $g_m$ (Fig. 4a). The equation indicates that for short-interval input spikes (e.g., $\Delta t < C_m/g_m$), an approximately linear temporal integration is expected, whereas for long-interval spikes (e.g., $\Delta t > C_m/g_m$) it will deviate to sub-linearity (bottom panel)[45]. Although excitatory postsynaptic current (EPSC) is different from the direct current injection modeled[46], close-to-linear and sub-linear postsynaptic temporal integrations were experimentally observed at high frequency (e.g., >200 Hz) and low frequency (e.g., <100 Hz) in biological neurons[45–47].

We hypothesize that the dynamics of filament formation in the memristor (Fig. 4b) is qualitatively analogous to the depolarization in a biological neuron (Fig. 4a). Specifically, the filament formation is also a 'threshold' event governed by the net Ag numbers ($N_{Ag}$) in a given filamentary volume ($S$), which corresponds to the net charges ($Q$) in a given cytosolic volume. A leaky term governed by $Ag^+$ diffusion (out of $S$) corresponds to the leaky current (out of cytosol). Based on Fick's law[48], the leaky term is proportional to $D \cdot \nabla\rho$, where $\rho$ is the $Ag^+$ concentration and $D$ the diffusion coefficient. As $Ag^+$ concentration outside the

filamentary volume is very low, $\nabla\rho$ is approximately proportional to $\rho$; so the leaky term can be written as $k \cdot \rho$ ($k$ is a coefficient). Therefore, a similar equation governing the state dynamics ($N_{Ag} = S \cdot \rho$) for filament formation can be expressed as $q \cdot S \frac{d\rho}{dt} = I_{Ag+} - k \cdot \rho$ (Eq. 2), where $q$ and $I_{Ag+}$ represent the charge quanta and $Ag^+$ injection current, respectively. Although the actual values of the components are determined by the physical properties of the protein nanowire-Ag system and largely unknown, the similar form between (Eq. 2) and (Eq. 1) indicates that similar temporal integration may be realized in the memristor to realize an artificial integrate-and-fire neuron.

We used input spikes (100 mV, 1 ms) close to biological action potential[9] and varied the frequency (or pulse interval) from 50 to 900 Hz to experimentally investigate the temporal integration in the artificial neuron (Fig. 4c). At a given frequency, the neuron integrated a certain number of spikes before its firing (turn-on) (Fig. 4d), followed by spontaneous repolarization or relaxation to HRS. The number of pulses featured a stochastic distribution (Fig. 4e), similar to that in a biological neuron[45]. Nonetheless, a clear difference in the average number of pulses was observed between high-frequency and low-frequency spikes (Fig. 4e). Collectively, the number of pulses needed for firing remained close in the high-frequency range (e.g., >200 Hz) but appeared to deviate (increased) in the low-frequency range (e.g., ≤100 Hz) (Fig. 4f). Such integration reflects a close-to-linear summation at high frequency and sub-linear one at low frequency, which is similar to the postsynaptic temporal integration by neurons[45–47].

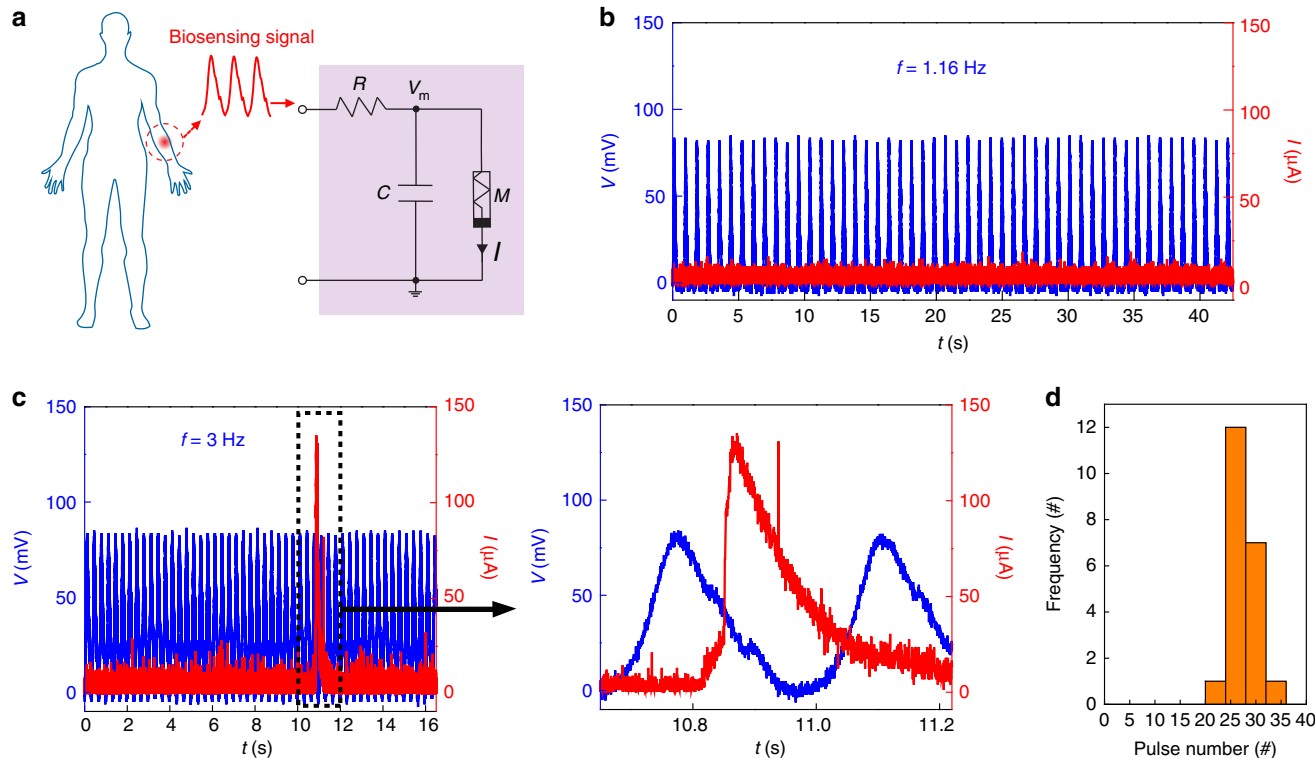

**Fig. 6 Bio-signal processing. a** Circuit of an artificial neuron constructed using bio-voltage memristor ($M$). A parallel capacitor $C = 100\,\mu F$ and series resistor $R = 10\,k\Omega$ were used to yield a time constant ($RC = 1\,s$). Emulated biosensing signals (e.g., pulse) were input to the neuron. **b** Pulse (blue) frequency of normal heart rate (1.16 Hz) could not trigger the neuronal firing (red). **c** Left panel, pulse (blue) frequency of abnormal heart rate (3 Hz) triggered the neuronal firing (red peak). Right panel, zoom-in pulse signal (blue) with dicrotic feature and triggered neuronal firing (red) featuring a fast depolarization and relatively slow repolarization similar to an action potential[52]. **d** Statistics of pulse number (26.3 ± 16.4, ±s.d.) needed to trigger neuronal firing at $f = 3\,Hz$.

In particular, it features a quantitative emulation as both frequency domains overlap with those in biological neurons[45].

**Bio-voltage artificial synapse**. In addition, the influx and efflux of Ag in the filamentary volume can also emulate $Ca^{2+}$ influx and extrusion in a biosynapse (Fig. 5a)[3,49]. The competing effect can lead to growth or shrinkage in the filament size, which is inferred from the experimental observation that a longer programming pulse yielded a longer relaxation (Supplementary Fig. 8). Consequently, the steady-state evolution in the built-up filament (conductance) can be used to emulate synaptic plasticity. We demonstrated a bio-voltage artificial synapse with a protein nanowire memristor (Fig. 5b). After activation (turn-on), the device's conductance increased with the increase in pulse number at high frequency (900 Hz), presumably because a high-rate $Ag^+$ injection into the filament volume would facilitate filament growth, similar to enhancement in synaptic strength by $Ca^{2+}$ surge[49]. Low-frequency (e.g., ~10 Hz) pulse train lead to conductance decrease (depression) (Fig. 5b), corresponding to a shrinkage in filament due to the dominance of Ag diffusion (leakage) over low-rate injection. Alternatively, tuning the pulse width at fixed frequency can also be used to modulate the conductance (Supplementary Fig. 21). These properties enabled the artificial synapse to realize frequency-dependent paired-pulse facilitation (PPF) and paired-pulse depression (PPD) (Fig. 5c), which implies the potential for autonomic computing. Compared with previous artificial synapses[3,13], the features of the protein-nanowire memristors are much closer to matching the parameters of biosynapses in signal amplitude (e.g., at least 10× smaller) and/or frequency range (e.g., <1 kHz).

**Bioelectronic interfacing**. Finally, we show the potential of implementing the bio-voltage memristors in biointerfaces. Various electronic devices such as self-powered wearable sensors[50,51] and intracellular bioprobes[52] have been developed to record physiological signals. The recorded signals are generally small and often in the sub-100 mV range[50-52], which require amplification before conventional signal processing. This pre-processing adds to the power and circuitry requirements for future closed-loop bioelectronic interfaces or biomimetic systems. The bio-voltage memristor provides the possibility for direct bio-signal processing to reduce the power and circuitry budget, which is highly desirable for improved sustainability and reduced invasiveness in bio-integrated systems.

Figure 6a shows the circuit of an artificial neuron with tunable integrate-and-fire response[14]. The input pulses gradually increase the voltage across the memristor ($V_m$) by charging up the capacitor ($C$) through the resistor ($R$). The equilibrium voltage peak is dependent on the input frequency relative to the time constant ($RC$). If the threshold voltage is reached, the memristor will be turned on and transits to low resistance ($R_{ON}$). If $R_{ON}$ is considerably smaller than $R$, it will discharge the capacitor to lower $V_m$. Meanwhile, the input pulses across the memristor will also be attenuated (e.g., by a factor of $R_{ON}/(R_{ON}+R)$ through the $R$-$R_{ON}$ voltage divider) and below the threshold. This forms a 'refractory' period similar to that in a biological neuron after firing. Therefore, the memristor naturally decays to Off after the firing, before it starts to integrate the next round of pulses. The input frequency-dependent firing in the artificial neuron can enable artificial bio-reporter to monitor bio-signal changes.

In a proof-of-concept demonstration, emulated biosensing signals (e.g., 80 mV pulses) were input to the artificial neuron

(Fig. 6a). The *RC* constant (~1 s) was tuned to be close to the period in normal heart rate, so the competing charging and discharging yielded a peak $V_m$ below the threshold and could not trigger the neuronal firing (Fig. 6b). Abnormal heart rate (e.g., 180 bpm) increased the charging rate to yield a $V_m$ larger than threshold and triggered the neuronal firing (Fig. 6c). Thus, the artificial neuron functioned as an on-site health reporter, showing the potential of bio-voltage memrisistors to perform on-site bio-signal processing. Cellular signal (e.g., action potential) has similar amplitude[16], although the retrieval of the intracellular amplitude requires an efficient cell-sensor coupling. Recent advances in 3D nanoscale bioprobes, including nanotransistors[52–54], nanopipettes[55,56], and nanopillar electrodes[57,58], have successfully demonstrated close-to-full-amplitude signal recording. These advances have indicated the potential in integrating intracellular bioprobes with bio-voltage memristors in new generations of bioelectronic interfaces, in which the bio-voltage artificial neuron may form direct communication with biological neurons.

**Bio-power computing**. Future bio-realistic systems (e.g., spiking neural network) are expected to integrate both synaptic (e.g., weighted interconnects) and somatic components (e.g., pulse-signal processing units). Previous nonvolatile memristors have been demonstrated to be an ideal candidate to realize synaptic functions[11]. Diffusive memristors, due to the inherent mimicry to an 'auto-reset' in biological signal processing (e.g., repolarization), can be advantageous in constructing the somatic pulse-processing components (e.g., artificial neurons). We anticipate that the complementary integration of the two types can lead to expanded neuromorphic functions, as were initially demonstrated in synaptic functions[3], capacitive networks[15], and spiking neural networks[14]. Moreover, the concept of introducing catalytic elements in memristors and the mechanism revealed can be employed to engineer bio-voltage nonvolatile memristors, hence possibly forming the complementary pair of volatile and non-volatile bio-voltage memristors for constructing bio-voltage neuromorphic systems. In an ideal spiking neural network (e.g., with minimal energy cost in the idle state), energy consumption will mainly go to the spiking events. As a result, reducing the spiking amplitude is expected to play the dominant role for power reduction. Our initial studies in single device (Supplementary Fig. 22) and neural network (Supplementary Fig. 23) have shown that the achieved bio-voltage amplitude can substantially reduce the energy cost in spiking events to close to or even below that of a biological neuron, demonstrating the potential in improving low-power neuromorphic computing.

## Discussion

In summary, we have demonstrated a new type of bio-voltage memristors. The low voltage can be attributed to the protein nanowires which facilitate cathodic $Ag^+$ reduction. Based on this, neuromorphic components such as artificial neuron and synapse functioning at biological action potential are realized. The artificial neuron has achieved the temporal integration similar to the frequency response in biological neurons. The bio-voltage operation substantially reduced the energy cost in constructed neuromorphic components (Supplementary Figs. 22 and 23). The protein nanowires are stable under harsh chemical and temperature conditions, providing broad options for the incorporation in electronic devices[27,59]. Similar to organic memristors[60–62], the protein composition, intrinsic flexibility and renewable production in the protein nanowires can also provide biocompatibility for biointerfacing. This biocompatibility, combined with the ability of the memristors to perform bio-parameter matched

computing, is expected to facilitate direct communication between electronic and biological interfaces. For example, bio-voltage memristors and neuromorphic components may be integrated in flexible substrates[62–64] for tissue interfaces, enabling on-site signal processing for close-loop bioelectronics. The catalytic concept may lead to broad efforts and generic strategies in modulating/reducing function voltages in various memristors.

## Methods

**Synthesis of protein nanowires**. The protein nanowires were harvested and purified from G. sulfurreducens as previously described[30]. Harvested nanowire preparation was dialyzed against deionized water to remove the buffer and stored at 4 °C. The resultant nanowire preparation yielded a measured pH~9.

### Device fabrication

*Planar nanogap devices*. A pair of contacts (*e.g.*, $80 \times 80 \, \mu m^2$ with extending interconnects), on which the probe tips land for electrical measurements, were first defined on a silicon substrate coated with 600 nm thermal oxide (Nova Electronic Mater., Inc.) through standard photolithography, metal evaporation (Au/Cr = 30/10 nm) and lift-off processes. Then standard electron-beam lithography (EBL) was employed to define a pair of Ag electrodes with nanogap configuration (e.g., 100-500 nm spacing), followed by metal evaporation (Ag/Ti = 200/3 nm) and lift-off processes (Supplementary Fig. 1). The protein nanowire solution was then drop-casted onto the device and thermally dried (e.g., 60–80 °C, 1–3 min) or naturally dried (e.g., 25 °C, 1 h) in the ambient environment. The drying rate did not affect the memristive result. The final nanowire-film thickness was controlled by tuning the solution volume over unit area. Empirically, 110 $\mu L/cm^2$ nanowire solution (150 $\mu g/mL$) yielded an average film thickness ~1 $\mu m$. For control devices (Supplementary Fig. 20) using polyvinylpyrrolidone (PVP) as the dielectric, PVP solution (3 mg/mL, molecular weight 360,000) was dropcasted onto the nanogaped electrodes and naturally dried in the ambient environment. The final film thickness was ~1.8 $\mu m$.

*Vertical devices*. The bottom electrode was first defined by standard lithography, metal evaporation (Pt/Ti = 20/3 nm) and lift-off processes on a silicon substrate coated with 600 nm thermal oxide. Then the top electrode pattern was defined by photolithography, followed by electron-beam evaporation of $SiO_2$ (25 nm) and Ag/Ti (150/3 nm) and lift-off processes. For the $100 \times 100 \, nm^2$ feature size (Supplementary Fig. 11), EBL was used to define the vertical device (electrodes were defined by photolithography). After the fabrication of the Pt–$SiO_2$–Ag device, the protein nanowire solution was dropcasted onto the device and naturally dried (e.g., 25 °C, 1 h) in the ambient environment. In a proof-of-concept demonstration to show the feasibility of patterning nanowire film in device fabrication (Supplementary Fig. 3), photolithography, dropcasting and lift-off processes were carried out to define patterned protein-nanowire film (~$50 \times 50 \, \mu m^2$) on to the device.

**Electrical measurements**. The electrical measurements were performed in the ambient environment, unless otherwise specified. The current-voltage ($I–V$) curves were measured by using an Agilent 4155C semiconductor parameter analyzer or a Keysight B1500A semiconductor analyzer. The pulsed measurements were carried out by using the B1530A waveform generator/fast measurement unit (WGFMU) integrated in the Keysight B1500A semiconductor analyzer. The emulated bio-sensing pulse signals were programmed using the WGFMU. The relative humidity (RH) in the ambient environment was real-time monitored by a hygrometer (Model 8706; REED Instruments). The higher RH (e.g., >50%) was controlled by tuning the equilibrium vapor pressure of sulfuric acid solutions[65] in a sealed testing stage containing the device (Supplementary Fig. 19).

**Material characterizations**. The thicknesses in protein nanowires and PVP films (cross section) were measured by a desktop scanning electron microscope (SEM, EM-30 Plus; Element Pi). The device structure and filamentary evolution were measured by a high-resolution SEM (JSM-7001F; JEOL). The nanowire-nanowire networks were imaged by using a transmission electron microscope (TEM, JEM-2200FS; JEOL). The $H_2O$ bonding spectra in protein nanowires were performed by a Fourier-transform infrared spectroscopy (FTIR; Perkin Elmer) equipped with a universal attenuated-total-reflection (ATR) sampling accessory.

**Cyclic voltammetry measurements**. The cyclic voltammetry was performed with a computer-controlled electrochemical analyzer (CHI 440 C, CH Instruments), using three-electrode configuration; Protein nanowires (~170 nm) or $SiO_2$ dielectric (~160 nm) were coated on Au electrodes (size ~25 $mm^2$) as the working electrodes. A platinum wire and Ag/AgCl electrode were used as the counter electrode and reference electrode, respectively. A 0.10 M $KNO_3$ solution (with $Ag^+$ 5 mM) was used as the electrolyte for the electrochemical measurements.

**Molecular dynamics simulation**. Detailed modeling protocols can be found in our previous work[30]. In brief, to examine the formation of water-permeable pores in protein nanowire networks, three nanowire-nanowire filaments, pili-pili, OmcS-OmcS, and OmcS-pili, were modeled using the latest CHARMM 36 m force field[66] implemented in the CHARMM program[67]. The initial structure of pili was obtained from a previous modeling work[68] and that of OmcS was from Protein Data Bank (ID: 6EF8). In all cases, one filament was set as the reference filament, with its principal axis oriented along x-axis. The other filament, in parallel with the reference filament, was mobile, which could both translate along y-axis and rotate along its filament axis (Supplementary Fig. 2a). These configurations represented possible tightest packing between filaments. For each packing configuration, the system was energy minimized and then equilibrated at 300 K for 3 ns. After simulation, the first half of each trajectory was discarded as the equilibration process, and the averaged potential energy was computed for each system. This allowed one to identify the most stable configuration, i.e., the inter-filament distance along y-axis and filament rotation with minimal averaged potential energy, for each shift distance along x-axis. The pore profiles were analyzed using the HOLE program[69] for these stable packing configurations, and radius of narrowest region along the pore profile was reported as the pore radius.

## Data availability

The data that support the findings of this study are available from the corresponding author upon reasonable request.

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

## Acknowledgements

J.Y. and D.R.L. acknowledge support from a seed fund through the Office of Technology Commercialization and Ventures at the University of Massachusetts, Amherst. J.Y. acknowledges the support by National Science Foundation (NSF) CBET-1844904. J.Y. acknowledge the helpful discussion from Prof. Qiangfei Xia. J.C. and X.R.L. acknowledge support from the National Institutes of Health (GM114300 to J.C.).

## Author contributions

J.Y. and T.F. conceived the project and designed experiments. D.R.L. oversaw material design and production. T.F. carried out experimental studies. X.M.L., H.G., B.Y., and D.J. F.W. helped material characterizations. J.E.W. performed material synthesis and imaging. X.R.L. and J.C. designed the computational study and analysis; X.R.L. performed simulations and analysis. Z.W., Y.Z., and J.J.Y. provided helpful discussion. All authors discussed the results and implications and commented on the manuscript at all stages.

## Competing interests

The authors declare no competing interests.
