## [Peer Review File · Nature Communications]

Reviewers' comments:

Reviewer #1 (Remarks to the Author):

This is an interesting concept with potential for leading to new applications and design of memristors but in my opinion it is not presented in a way that is convincing and therefore I recommend major revision.

Arguments for this opinion are found below:

The structure of OmcS protein, that forms conductive filaments in *Geobacter* was recently published in *Cell* and is not mentioned. Indeed the molecular modeling used by the authors reported in supplementary figure 2 is based on pilA whose capacity to conduct electrons remains controversial. Furthermore, in the materials and methods section, synthesis and purification of protein nanowires, the authors report no attempt to unequivocally identify the molecular nature of the biological material used. For instance MS would be a trivial measurement that would identify peptides that could be matched to proteins in the genome.

The use of acetone in one of the steps for fabrication of the memristome as reported in supplementary figure 3 leads me to question the structural integrity of the protein after the treatment and thus the relevance of the molecular modelling reported in figure 2.

The fact that as reported in the materials and methods section, fabrication of memristive devices, the authors report that drying at high temperature leads to material with properties similar to drying at room temperature again leads me to question the structural integrity of the biological material deposited, and I do not see in the description of the work evidence that the authors controlled this issue.

Minor issues that I found are:

Figure 4 panel f: does the dashed line represent a mathematical model of the phenomenon or is it there as a visual guide. If it is the second option, it should be made of linear segments connecting the experimental points, as the authors used in panel c of figures 3 and 5, for example.

The line in supplementary figure 8 does not pass through the origin (0,0) which I find unlikely to be physically sensible because, as it stands, it would mean negative retention time for pulses shorter than ~ 10 ms!

Finally, there appears to have been little care in preparation of text with multiple typos or imprecisions: for example p1, line 71 *G sulfureducens* is a bacterium. Referring to it as 'organism' is unnecessarily imprecise.

Reviewer #2 (Remarks to the Author):

In this work the authors show that a memristor based on protein nanowires can operate with low voltages near the biological potential of 40-100 mV. This potentially reduces power and can be implemented in direct communication with biology.

The paper is well written and reads well. Figures are clear and the story is to the point. However, to warrant publication I do have some comments to be addressed.

My main concern is the inherent decrease in stability by decreasing the switching voltage. To my knowledge a low switching (writing) voltage will also mean that by probing (reading) the device, the conductance state can be (slightly) modified. There is a tradeoff between read and write energy and stability of the state, specifically when introducing multiple conductance states. This is

also apparent when considering the mechanism of the leaky-integrate and fire neuron demonstrated in Fig 4b. Can the authors comment on this?

Another comment is regarding the claim that these low voltage devices open up a path for directly communicative electro-bio interfaces. If this is one of the main applications (apart from lower power consumption), can the authors further comment on the toxicity of silver in bio applications as well as stability of these devices in a biological environment?

The second part of the paper introduces a neuron functionality similar to a Leaky-Integrate-and-Fire. This is an interesting concept and perhaps I am misreading this part, but if the neuron device is not activated anymore, will the device go back to its original state as a real neuron would do? But does that not imply the new conductance state has a low stability? And how does that relate to the memristor state stability?

Furthermore, if the protein nanowires facilitates the cathodic reduction of Ag^+ , does that mean that only tuning the conductance one way is improved? Or how does this translate to conductance tuning the other way? From the graph (Fig 2b) it looks like both up and down tuning initiates at low voltages.

Reviewer #3 (Remarks to the Author):

Review of manuscript for Nat. Comm.
Bioinspired Bio-voltage Memristors
Tianda Fu, et al., (U Mass Amherst).

The authors present an experimental demonstration of a novel memristor device based on ionic silver switches whose formation is mediated by protein nanowires. These devices are shown to have a low operation voltage that is around 100 mV. These voltages are similar to the voltages that occur in biological systems. The demonstration of memristive devices operating as such low-voltages is impressive and is the notable result in this report.

However, this demonstration alone is not a significant enough advancement to merit publication in a high-profile journal such as Nature Communications.

The authors do not provide sufficiently compelling experimental support for their proposed mechanism. The authors propose a three-step mechanism dominated by cathodic reduction for the switching observed in these devices, and they show that protein nanowires can be key to lowering the switching voltage. However, they do not unambiguously demonstrate that a change in the electrochemical behavior of the system is lowering the switching. Figure 3d shows cyclic voltammetry for the Ag on the wire system, but there is effectively no electro chemistry occurring on the SiO₂ only samples; thus, the "shift" is not convincing. In addition, is it necessary to use protein nanowires? or is it simply an effect of having transport matrix present? Would a CNT matrix or network of other nanowires work just as well? A tremendous amount of Ag is involved in the switching, does changing the surface potential of the SiO₂ with a surfactant change the switching voltage?

More critical in my mind for showing a significant advancement would be a demonstration of an end application that requires the low switching voltage. For example, there is no demonstration of a direct connection to a biological system or an ultra-low powered neuromorphic circuit implementing a useful search or visualization operation. There is not even a demonstration that these memristors can be scaled to large systems. Without a demonstration of an impactful application that is enabled by these low-voltage devices, the paper does not make the impact necessary to warrant publication in Nature Communications.

Please note, that I find no major technical flaws in the material that is presented, and I suggest that the paper is submitted for publication to a more technically focused journal.

Following are some minor features of the paper that when improved will make the paper easier to understand and will help it have a bigger impact on the community.

There are many small grammatical errors throughout the paper. The readability of the paper would be greatly improved if a thorough edit was performed with an eye towards making it grammatically correct. By no means am I going to illustrate all the errors, but I'll show a couple of examples here:

1st paragraph: "... there has been emerging attempts to use memristors ..." ("has" should be "have" and attempts don't emerge: "... there have been attempts to use memristors ... "

Page 6, 2nd paragraph. This is an important paragraph in the paper, and very challenging to understand as written due to a combination of poor grammar, challenging concepts, and poor ordering of the introduction of concepts. In particular.

"As neural firing is triggered by membrane potential (V_m) directly related to net cytosolic charge ($Q = C_m \cdot V_m$; C_m is the membrane capacitance), the state dynamics is often modeled⁴³ by $C_m \frac{dV_m}{dt} = I - g_m V_m$ (Eq. 1), where I denotes injection current and $g_m V_m$ the leaky current related to the membrane conductance g_m (Fig. 4a)."

Q is not defined. If I understood this complex sentence properly, one way that could be written is as follows.

Neural firing is triggered by the membrane potential (V_m) which is directly related to the net charges (Q) in a given cytosolic volume ($Q = C_m \cdot V_m$; C_m is the membrane capacitance); therefore the state dynamics are often modeled⁴³ by $C_m \frac{dV_m}{dt} = I - g_m V_m$ (Eq. 1), where I denotes injection current and $g_m V_m$ the leaky current related to the membrane conductance g_m (Fig. 4a).

As this paper is primarily concerning electronic device behavior, it would probably be helpful to explain cytosolic. The paragraph below the one containing the above sentence does explain the hypothesis "that the dynamics of filament formation in the memristor (Fig. 4b) is qualitatively analogous to the depolarization in a biological neuron". But the explanation comes after the initial paragraph that creates the confusion.

Because both cyclic voltammetry (CV) and capacitance (C) are referred to in this paper, it is important that the authors always make it clear which they are referring to. I suggest that they write out the words cyclic voltammetry in the figure caption to Fig. 3.

I don't see where Figures 4A and 4B are referred to in the body of the text. What is their purpose?

Be sure that the supplemental figures have complete figure captions or accompanying notes that ensure that the presented data and its technical importance can be fully understood. Following are a few issues that were easily noticed.

SFig 4. & 5 & 7 & 8. &17. Be sure to indicate what device structure (planar or vertical) is used for these data.

- 4 - protein-nanowire device (nothing indicated geometry)
- 5 - protein-nanowire memristor (nothing indicated geometry)
- 6 - protein-nanowire memristors (picture indicates geometry)
- 7 - protein-nanowire memristor (nothing indicates geometry)
- 8 - protein-nanowire device (nothing indicates geometry)

9 - vertical protein-nanowire memristors (name and picture indicate geometry)

17 - protein-nanowire memristor (also protein-nanowire synapse) (nothing indicates geometry)

Supplementary Fig 10. Right. The vertical devices are effectively two devices in parallel. What is the appropriate scaling factor? (It doesn't matter here because they are all treated similarly, but what is the appropriate length for comparing vertical to horizontal devices?)

What is the difference in the conditions for SFig. 11 and SFig13.?

SFig. 16 (b) What does the data represent? Is it the initial formation switch for five separate devices? Is it the first five cycles of a single device?

Response to Reviewers

To make it clear, we have used *italic* fonts for the reviewers' comments, **black** fonts for our replies, and **blue** fonts for revisions.

Reviewer #1 (Remarks to the Author):

This is an interesting concept with potential for leading to new applications and design of memristors but in my opinion it is not presented in a way that is convincing and therefore I recommend major revision.

We greatly thank the reviewer for confirming the novelty in the work, both from the concept and potential application perspectives. We also thank the reviewer for the suggestions/comments to strengthen the scientific presentation. Below please find our detailed responses to the reviewer's questions.

The structure of OmcS protein, that forms conductive filaments in Geobacter was recently published in Cell and is not mentioned. Indeed the molecular modeling used by the authors reported in supplementary figure 2 is based on pilA whose capacity to conduct electrons remains controversial. Furthermore, in the materials and methods section, synthesis and purification of protein nanowires, the authors report no attempt to unequivocally identify the molecular nature of the biological material used. For instance, MS would be a trivial measurement that would identify peptides that could be matched to proteins in the genome.

We appreciate the reviewer's update on the recent development in protein nanowires. We are well aware of the formation of OmcS filaments as one of the authors of our manuscript was also an author on the first publication of the OmcS filament structure (<https://www.biorxiv.org/content/10.1101/492645v1.abstract>). However, as recently reviewed in detail (*Front. Microbiol.* 2019, 10, 2078), the abundance of PilA-based filaments and OmcS-based filaments is highly dependent upon culture conditions. In the studies reported here the methods employed yielded filament preparations comprised primarily of PilA-based filaments. In contrast to the reviewer's comment, mass spectrometry (MS) is not a "trivial method" when identifying PilA-based filaments because the filaments are not denatured to subunits and the PilA subunit does not 'fly' in typical MS methods. However, it is possible to distinguish between PilA-based filaments and OmcS-based filaments based on the differences in their diameters. We demonstrate here that the diameters of the filaments employed were 3 nm, consistent with the diameter of PilA-based filaments and inconsistent with the 4 nm diameter of OmcS-based filaments (Fig. R1).

What is more relevant to this study is whether the coexistence of OmcS nanowire plays a role in the memristive effect. Therefore, we have done control study by using nanowires harvested from a strain of *Geobacter* in which the gene for OmcS was deleted. In the revised

manuscript, we demonstrate that protein nanowires obtained from a mutant strain of *Geobacter* that could not produce OmcS functioned as well in memristor devices (Fig. 2R) as protein nanowires derived from wild-type *Geobacter*. These results demonstrate that PilA-based filaments rather than OmcS-based filaments were responsible for memristor function.

It is also important to recognize that the co-existence of PilA-based and OmcS-based nanowires would not significantly alter the structure of the protein nanowire films.

For example, we have performed further molecular dynamics simulations by including OmcS nanowires. As both pilA and OmcS nanowires have helical structures, their combinations still yield molecular pores at the nanowire-nanowire interface (Fig. R3).

So in the revised manuscript, we have (1) added the possibility of the co-existence of OmcS nanowires by citing latest studies including the *Cell* paper, (2) extended the molecular dynamics simulations (e.g., adding Fig. R3 to Supplementary Fig. 3), and (3) discussed that OmcS nanowire is not likely to be the contributing factor, by adding the new results of Figs. R1 & 2 to a new Supplementary Fig. 1e and Supplementary Fig. 2) and the description in main text (page 4):

“...Two types of protein nanowires can be recovered from *Geobacter sulfurreducens*, protein wires that assemble from the pilin monomer PilA and wires that assemble from the c-type cytochrome OmcS.²⁸ The relative abundance of each type of wires depends upon the conditions under which the cells are grown. The protein nanowires in our preparations (Supplementary Fig. 1) had an average diameter of 2.9 ± 0.35 nm, which is consistent with the 3 nm diameter of pilin-based nanowires²⁸ and inconsistent with the 4 nm diameter of protein nanowires comprised of OmcS.^{30,31} Furthermore, as detailed below, devices constructed with protein nanowires harvested from a strain of *Geobacter sulfurreducens* in which the gene for OmcS was deleted yielded similar results (Supplementary Fig. 2). These results suggested that pilin-based protein nanowires were the important functional components.”

The use of acetone in one of the steps for fabrication of the memristor as reported in supplementary figure 3 leads me to question the structural integrity of the protein after the treatment and thus the relevance of the molecular modelling reported in figure 2.

The fact that as reported in the materials and methods section, fabrication of memristive devices, the authors report that drying at high temperature leads to material with properties similar to drying at room temperature again leads me to question the structural integrity of the biological material deposited, and I do not see in the description of the work evidence that the authors controlled this issue.

We thank the reviewer for the careful thought over the structural integrity in material after different treatments. Previous studies have shown that nanowires from *Geobacter* are stable in harsh environment (e.g., pH 2-10), against organic solvents (e.g., detergent), and at high temperature (e.g., 100 °C) (*Curr. Opin. Electrochem.* 2017, 4, 190; *mBio* 2017, 8, e00695-17). Such

Fig. R2. Sub-100 mV memristive switching in a device made from nanowires harvested from *Geobacter* with OmcS gene deletion.

Fig. R3. MD simulations of pilA-pilA, OmcS-OmcS, and OmcS-pilA nanowire interfaces (left), with corresponding range in the interfacial pore size (right).

stability is only found in some proteins, an analogous example is amyloid proteins. This can be a remind of nature's design to allow certain organisms to live in different environments (e.g., algae/bacteria in high-temperature environments (*Science* 1967, 158, 1012)).

To further address the reviewer's concern, we have done additional studies. First, the same nanowire film prepared at room temperature (RT) showed negligible change in conductance after it was heated to different temperatures of 50°, 70°, 90°, 110° and cooled down to RT (Fig. R4). Drop-casting acetone on the film yielded some conductance decrease but maintained the same level (yellow line). This conductance change can come from the sensing effect to acetone adsorption, which has been constantly observed in other thin-film devices/sensors (*Nanomaterials* 2017, 7, 339).

Fig. R4. Current-voltage curves (IVs) from the same protein-nanowire film after different treatments.

More importantly, we have also tested the memristive switching in three devices with the nanowires prepared by drying at (1) RT, (2) 110°, and (3) RT with acetone cleaning, respectively. The three devices showed consistent bio-voltage memristive switching (Fig. R5).

Fig. R5. Sub-100 mV switching in three devices with the nanowire film prepared in different conditions.

We believe that the prior studies and our new controls are convincing evidence that the nanowires can withstand the processing conditions for device application, although we acknowledge that current processes are still in the proof-of-concept demonstration stage. In the revised manuscript, we have cited the prior studies to indicate the stability in nanowires (page 8): “...The protein nanowires are stable under harsh chemical and temperature conditions, providing broad options for the incorporation in electronic devices.^{27,55}”

Figure 4 panel f: does the dashed line represent a mathematical model of the phenomenon or is it there as a visual guide. If it is the second option, it should be made of linear segments connecting the experimental points, as the authors used in panel c of figures 3 and 5, for example.

We thank the reviewer for this careful check. This is not a fitting of any mathematical model. Therefore, we have changed to linear segments as the reviewer suggested in the revised manuscript.

The line in supplementary figure 8 does not pass through the origin (0,0) which I find unlikely to be physically sensible because, as it stands, it would mean negative retention time for pulses shorter than ~10 ms!

We thank the reviewer for this careful observation. Pulse width less than 10 ms simply

cannot turn on the device (Fig. 2d, reproduced as Fig. R6a below). This means that for $t \leq 10$ ms, the retention is essentially *zero* and flat. In the linear fitting, we did not include this flat region. In fact, the fitting yielded an extrapolated *zero* retention at ~ 10 ms input pulse width (Fig. R6b), which is expectedly consistent with experimental result.

To avoid confusion, we have added following explanation in Supplementary Figure 8 (now FigS9): “...Note that this linear fitting does not pass the origin, because only pulses with width >10 ms can turn on the device (Fig. 2d). The extrapolated 0 ms retention at ~ 10 ms input pulse width is consistent to the experimental result (Fig. 2d)”

Finally, there appears to have been little care in preparation of text with multiple typos or imprecisions: for example, p1, line 71 *G sulfureducens* is a bacterium. Referring to it as 'organism' is unnecessarily imprecise.

We have replaced the word 'microorganism' with 'bacterium'. We now have thoroughly checked the languages in the manuscript to ensure a precise presentation (the reviewer can tell our efforts in a Tracked version uploaded together). We thank the reviewer for the time and helpful suggestions. We also wish the reviewer a happy holiday.

Reviewer #2 (Remarks to the Author):

In this work the authors show that a memristor based on protein nanowires can operate with low voltages near the biological potential of 40-100 mV. This potentially reduces power and can be implemented in direct communication with biology.

The paper is well written and reads well. Figures are clear and the story is to the point. However, to warrant publication I do have some comments to be addressed.

We greatly thank the reviewer for confirming the value/potential of the research and the quality of the scientific presentation. We also appreciate the reviewer's suggestions/comments to help strengthen the work. Below please find our detailed responses to the questions.

My main concern is the inherent decrease in stability by decreasing the switching voltage. To my knowledge a low switching (writing) voltage will also mean that by probing (reading) the device, the conductance state can be (slightly) modified. There is a tradeoff between read and write energy and stability of the state, specifically when introducing multiple conductance states. This is also apparent when considering the mechanism of the leaky-integrate and fire neuron demonstrated in Fig 4b. Can the authors comment on this?

We appreciate the reviewer's general view over memristor stability. Please allow us to start with some general perspectives before getting to the specific study here.

Mechanistically, memristors can be classified into electrochemical metallization cells (EMC) and valence change memory (VCM) devices. The memristors that fit the mathematical model were generally VCM devices, in which the analogous conductance change can be described by the continuous drift of oxygen vacancies (*Nature* 2008, 453, 80). In such model, any signal input in principle can cause the drift in vacancy thus the conductance change. In a real device, a low activation energy is still associated with the oxygen vacancy (*Sci. Rep.* 2019, 9, 17019), so a sufficiently low reading voltage (*e.g.*, smaller than the activation energy) is expected to cause negligible state change. But it is a legitimate concern that a reduced writing voltage generally means a reduced activation energy, so the reading voltage has the increasing chance to perturb the written state.

The EMC memristor, on the other hand, is more of a 'threshold' device featuring abrupt conductance change that corresponds to the physical rupture or re-bridging of a metal filament. This also means that static multiple conductance states are less feasible in EMC devices. A reading voltage below the 'threshold' voltage can hardly change the conductance. This is because even though the filament formation is based on field-driven ion migration, the cation ions are not readily available (*v.s.* the readily available oxygen vacancy in VCM). Instead it requires electrochemical redox process (*i.e.*, metal oxidization/reduction), which is largely a threshold event by overcoming the electrochemical reaction potential (*Appl. Catal. B* 2017, 202, 217). This has been experimentally shown in other study that even though the switching voltage in EMC was low (*e.g.*, ~0.2 V), the continuous reading did not alter the conductance state (*Nano Lett.* 2019, 19, 2411).

Our device falls into the EMC category, more precisely, is a volatile EMC memristor. The continuous conductance change in the device is a dynamic modulation that only happens with writing voltage pulses (*e.g.*, Fig. 4d, Fig. 5b), because the dynamic filament evolution still requires a threshold 'reduction' step ($\text{Ag}^+ \rightarrow \text{Ag}$) as illustrated in Fig. 4b. A reading voltage smaller than the writing voltage cannot overcome the reduction overpotential, and hence is not expected to alter the dynamic conductance state.

Another comment is regarding the claim that these low voltage devices open up a path for directly communicative electro-bio interfaces. If this is one of the main applications (apart from lower power consumption), can the authors further comment on the toxicity of silver in bio applications as well as stability of these devices in a biological environment?

We thank the reviewer for a visionary thought over the future potential. This concern can be viewed from two perspectives. First, Ag is applied in practical medicine such as eye treatment and treatment of skin ulcers (*Int. J. Res. Pharm. Sci.* 2017, 4, 1). It is also used as antibacterial material for wound treatment and surgical instruments (*J. Biomed. Mater. Res. A* 2012, 100, 1033). Of course, excessive Ag absorption can cause organ disruption and potential diseases. The memristor contains minimal trace of Ag (e.g., $\sim\mu\text{g}$ in 1 million devices) that is far from biosafety concern. In fact, it is more of a realistic concern from device perspective that, if Ag were dissolved, whether the memristor would still maintain the function.

Both concerns can be addressed by sealing the device body while only exposing the terminals to interface biological tissue for signal retrieval and feedback. In this case, the interconnects or terminals no longer need to be made of Ag, but can be of inert metals such as Pt/Au shown to be biocompatible. There are mature technologies to integrate electronics in sealed flexible substrates for tissue interface (*Nano Lett.* 2017, 17, 5836). For example, there is study showing that Ag memristors were integrated in a skin interface and maintained the functionality using such strategy (*Nat. Nanotechnol.* 2014, 9, 397).

In the revised manuscript (page 8), we now include a brief discussion of the potential strategy for electro-bio interface: "...For example, bio-voltage memristors and neuromorphic components may be integrated in flexible substrates^{53,54} for tissue interfaces, enabling on-site signal processing for close-loop bioelectronics."

The second part of the paper introduces a neuron functionality similar to a Leaky-Integrate-and-Fire. This is an interesting concept and perhaps I am misreading this part, but if the neuron device is not activated anymore, will the device go back to its original state as a real neuron would do? But does that not imply the new conductance state has a low stability? And how does that relate to the memristor state stability.

We think that there is little misunderstanding here. The memristor described here is a volatile one, as it naturally decays to Off if the writing input is removed (Fig. 2d). Such type of memristors are also referred to as 'diffusive' memristors and shown to be important in encoding the relative temporal information in neuromorphic systems (*Nat. Mater.* 2017, 16, 101). In this regard, the memristor naturally goes back to a low-conductance or 'rest' state after the firing, just like the repolarization process in a real neuron.

Fig. R1. (a) Circuit diagram of an artificial neuron. (b) Integrate-and-fire in the artificial neuron (blue spike), after which the neuron naturally decays to a 'rest' state.

A closer version is to add a series resistor (R) and a parallel capacitor (C) to facilitate the

‘repolarization’ process (Fig. R1a). When the memristor is turned on, it quickly discharges the capacitor to bring down the terminal voltage (V_m). At the same time, the series resistor R and the low-resistance memristor (R_{ON}) form a voltage divider, so input pulse is attenuated across the memristor (e.g., by a factor of $R_{ON}/(R_{ON}+R)$) and below the writing threshold. This forms a period like the ‘refractory’ period in a biological neuron after firing. Therefore, the memristor naturally decays to Off after the ‘firing’ (Fig. R1b), before it begins to integrate the next round of pulses—just like a biological neuron.

In the revised manuscript, we now **have added this new version of artificial neuron (Fig. R1) to a new Fig. 6 with corresponding (above) descriptions (Page 7-8).**

Furthermore, if the protein nanowires facilitate the cathodic reduction of Ag^+ , does that mean that only tuning the conductance one way is improved? Or how does this translate to conductance tuning the other way? From the graph (Fig 2b) it looks like both up and down tuning initiates at low voltages.

Theoretically, the electrochemical process in a writing/set process ($Ag \rightarrow Ag^+ \rightarrow Ag$) is the same as the one in an erasing/reset process ($Ag \rightarrow Ag^+ \rightarrow Ag$), with the former moving Ag from electrode to filament (turn on) and the latter moving Ag from filament to electrode (turn off). If the cathodic reduction is the determining step and nanowire facilitates the step, we would still expect a same facilitation in the reset process due to the symmetry (note that the cathode and anode also swap in the two processes).

In reality, as mentioned before, the memristor described here is a volatile one. The turn-off process is not driven by electrical input, but by the interfacial energy relaxation in the filament (*Nat. Mater.* 2017, 16, 101), i.e., through a natural decay (Fig. 2d). Fig. 2b essentially describes (only) the turn-on processes in both polarities (due to a symmetrical planar device structure). As a result, the cathodic facilitation from protein nanowire in principle would work in both processes, whereas here only needs to work in the turn-on process.

We thank the reviewer for the time and helpful suggestions. We also wish the reviewer a happy holiday.

Reviewer #3 (Remarks to the Author):

The authors present an experimental demonstration of a novel memristor device based on ionic silver switches whose formation is mediated by protein nanowires. These devices are shown to have a low operation voltage that is around 100 mV. These voltages are similar to the voltages that occur in biological systems. The demonstration of memristive devices operating as such low-voltages is impressive and is the notable result in this report.

However, this demonstration alone is not a significant enough advancement to merit publication in a high-profile journal such as *Nature Communications*.

We thank the reviewer for confirming the device novelty and the impressive result in sub-100 mV function. We believe that the research has also provided a concept change in the field and vocalized (i) that neuromorphic emulation can be beyond the functional level and toward bio-parameter matching, and (ii) a catalytic concept for broad engineering strategy. These ideas can lead to new frontiers in pursuing bio-voltage electronics or neuromorphic-bio interfaces. And we believe that such voice and impact live up to *Nature Communication* standard (as was also inferred from the other reviewers).

We do take the reviewer's points seriously and have further demonstrated the unique potential in applications. The revised version now shows both novelty in device and impact in potential applications, providing a more strengthened and comprehensive study for *Nature Communication* standard. Below please find our detailed responses to the questions.

The authors do not provide sufficiently compelling experimental support for their proposed mechanism. The authors propose a three-step mechanism dominated by cathodic reduction for the switching observed in these devices, and they show that protein nanowires can be key to lowering the switching voltage. However, they do not unambiguously demonstrate that a change in the electrochemical behavior of the system is lowering the switching. Figure 3d shows cyclic voltammetry for the Ag on the wire system, but there is effectively no electro chemistry occurring on the SiO₂ only samples; thus, the "shift" is not convincing.

We believe it is because of the less clear plot in Fig 3d (e.g., diminished peaks due to scaling) that gave the impression that 'there is effectively no electrochemistry occurring on SiO₂ only samples'. We have now replotted the curves in different scales (Fig. R1), which we believe can now clearly show the 'shift'.

We provide some explanation to the evidence. In electrochemistry, it is standard to coat different materials on electrode to study the catalytic effect. Both the shifts in the redox peak (*Chem* 2019, 5, 2429) and starting position of current increase (*Adv. Mater.* 2018, 30, 1707319) are used to show facilitation. The new figure shows no shift/facilitation in oxidation peak (gray dashed line), but a shift/facilitation in reduction. The less apparent reduction peak in SiO₂ was consistent to other SiO₂-coated cyclic voltammetry curves (*Coating* 2019, 9, 487; *Nano Biomed. Eng.* 2018, 10, 156). So here we used the shift in the starting position of current increase as the indication (black dashed line).

In the revised manuscript, we now have replotted Fig. 3d (as shown in Fig. R1 here) to show a clear shift. We thank the reviewer's help in improving the clearness in presentation.

Fig. R1. Cyclic voltammetry using nanowire-coated and SiO₂-coated electrodes. The arrow shows the shift in the starting position of current increase.

In addition, is it necessary to use protein nanowires? or is it simply an effect of having transport matrix present? Would a CNT matrix or network of other nanowires work just as well?

We thank the reviewer for bringing about other possible controls to further confirm the protein-nanowire's unique role.

We now have performed controls using both single-walled CNT matrix and Si-nanowire network. Specifically, devices using CNT-matrix initially showed high-conductance transport dominated by CNT (Fig. R2a), whereas devices using semiconducting Si-nanowire network showed low-conduction like bare SiO₂ dielectric (Fig. R2b). None of them formed low-voltage switching following the same forming process. The results are consistent with previous studies (*App. Phys. Lett.* **2017**, 111, 153504; *J. Appl. Phys.* **2018**, 124, 152118), in which Ag memristors using nanowire mesh/network did not yield low-voltage switching (*e.g.*, >1 V). These 'negative' controls indicate that it is not the percolation material structure that contributes to the effect, which is a further indirect support to the catalytic effect from the protein nanowires.

Fig. R2. IVs from devices using (a) CNT and (b) Si-nanowire network.

In the revised manuscript, we have now added these results to a new Supplementary Fig. 17 as negative controls to support protein-nanowire's unique role.

A tremendous amount of Ag is involved in the switching, does changing the surface potential of the SiO₂ with a surfactant change the switching voltage?

The initial forming may have consumed good amount of Ag, which nonetheless is typically observed in other Ag-based memristors (*Adv. Funct. Mater.* **2014**, 24, 5679; *J. Appl. Phys.* **1976**, 47, 2767). We thank the reviewer for the careful thought that the Ag filament may contact SiO₂ surface and hence the surface property of SiO₂ might have contributed to the bio-voltage switching.

We therefore performed controls in which the SiO₂ surface was functionalized with different surface potential. We used both anionic (α -NH₂, ω -COOH-terminated polyethylene glycol) and cationic (ω -Amino-terminated poly(ethylene glycol) methyl ether) surfactants to change the surface charge states and hence surface potential in SiO₂ before depositing protein nanowires. Both devices showed the same bio-voltage switching as protein-nanowire devices made on pristine SiO₂ (Fig. R3). SiO₂ surface functionalization alone (without protein nanowire) could not yield bio-voltage memristive switching.

Fig. R3. Bio-voltage switching in devices with SiO₂ functionalized with (a) anionic and (b) cationic surfactants.

We believe that this is another set of controls to show the contributing role from the protein nanowires. In the revised manuscript, we now added Fig. R3 to a new Supplementary Fig. 16 as further support to nanowire's role.

More critical in my mind for showing a significant advancement would be a demonstration of an end application that requires the low switching voltage. For example, there is no demonstration of a direct connection to a biological system or an ultra-low powered neuromorphic circuit implementing a useful search or visualization operation. There is not even a demonstration that these memristors can be scaled to large systems. Without a demonstration of an impactful application that is enabled by these low-voltage devices, the paper does not make the impact necessary to warrant publication in *Nature Communications*.

We thank the reviewer for the critical thought. While the work focuses on device, we now have also demonstrated the potential in implementing the bio-voltage memristors as described as follows (page 7-8):

“Finally, we show the potential of implementing the bio-voltage memristors in biointerfaces. Various electronic devices such as intracellular bioprobes⁵⁰ and self-powered wearable sensors^{51,52} have been developed to record physiological signals. The recorded signals are generally small and often in the sub-100 mV range,⁵⁰⁻⁵² which require amplification before conventional signal processing. This pre-processing adds to the power and circuitry requirements for future closed-loop bioelectronic interfaces or biomimetic systems. The bio-voltage memristor provides the possibility for direct bio-signal processing to reduce the power and circuitry budget, which is highly desirable for improved sustainability and reduced invasiveness in bio-integrated systems.

Fig. R4 (Fig. 6). **a**, Circuit of an artificial neuron constructed using bio-voltage memristor (M). $C=100\ \mu\text{F}$, $R=10\ \text{k}\Omega$ were used to yield a time constant ($RC=1\ \text{s}$). Emulated biosensing signals (e.g., pulse) were input to the neuron. **b**, Pulse (blue) frequency of normal heart rate (1.16 Hz) could not trigger the neuron firing (red). **c**, Left panel, pulse (blue) frequency of abnormal heart rate (3 Hz) triggered the neuron firing (red peak). Right panel, zoom-in pulse signal (blue) with dicrotic feature and triggered neuron firing (red) featuring a fast depolarization and relatively slow repolarization similar to an action potential. **d**, Statistical pulse number (26.3 ± 16.4) needed to trigger neuron firing.

Fig. 6a shows the circuit of an artificial neuron with tunable integrate-and-fire response.¹⁴ The input pulses gradually increase the voltage across the memristor (V_m) by charging the capacitor (C) through the resistor (R). The equilibrium voltage peak is dependent on the input frequency relative to the time constant (RC). If a threshold voltage is reached, the memristor will be turned on and transits to low resistance (R_{ON}). If R_{ON} is considerably smaller than R, it will discharge the capacitor to lower V_m . Meanwhile, the input pulses across the memristor will also be attenuated (e.g., by a factor of $R_{ON}/(R_{ON}+R)$ through the R- R_{ON} voltage divider) and below the threshold. This forms a ‘refractory’ period similar to that in a biological neuron after firing. Therefore, the memristor naturally decays to Off after the firing, before it starts to integrate the next round of pulses. The frequency-dependent firing in the artificial neuron can enable artificial bio-reporter to monitor bio-signal changes.

In a proof-of-concept demonstration, emulated biosensing signals⁵⁰⁻⁵² (e.g., 80 mV pulses) were input to the artificial neuron (Fig. 6a). The RC constant (~1 s) was tuned to be close to the period in

normal heart rate (e.g., $f=1.16$ Hz), so the competing charging and discharging yielded a peak V_m below the threshold and could not trigger the neuron firing (Fig. 6b). Abnormal heart rate (e.g., $f=3$ Hz) increased the charging rate to yield a V_m larger than threshold and triggered the neuron firing (Fig. 6c). The firing showed a stochastic feature (Fig. 6d). The artificial neuron here hence realized an on-site health reporter, showing the potential of using bio-voltage memristors to do direct bio-signal processing. Cellular signal (e.g., action potential) has similar amplitude,⁵⁰ and thus the bio-voltage memristor may also enable artificial interneuron for direct cellular communication.”

Please note, that I find no major technical flaws in the material that is presented, and I suggest that the paper is submitted for publication to a more technically focused journal.

We thank the reviewer for confirming the solid research. We believe that the device novelty, now combined with above demonstrated potential in applications, shows the broad impact suitable for *Nature Communications*.

Following are some minor features of the paper that when improved will make the paper easier to understand and will help it have a bigger impact on the community.

There are many small grammatical errors throughout the paper. The readability of the paper would be greatly improved if a thorough edit was performed with an eye towards making it grammatically correct. By no means am I going to illustrate all the errors, but I'll show a couple of examples here:

1st paragraph: “... there has been emerging attempts to use memristors ...” (“has” should be “have” and attempts don't emerge: “... there have been attempts to use memristors ...”

Page 6, 2nd paragraph. This is an important paragraph in the paper, and very challenging to understand as written due to a combination of poor grammar, challenging concepts, and poor ordering of the introduction of concepts. In particular.

“As neural firing is triggered by membrane potential (V_m) directly related to net cytosolic charge ($Q = C_m \cdot V_m$; C_m is the membrane capacitance), the state dynamics is often modeled by $C_m \frac{dV_m}{dt} = I - g_m V_m$ (Eq. 1), where I denotes injection current and $g_m V_m$ the leaky current related to the membrane conductance g_m (Fig. 4a).”

Q is not defined. If I understood this complex sentence properly, one way that could be written is as follows.

Neural firing is triggered by the membrane potential (V_m) which is directly related to the net charges (Q) in a given cytosolic volume ($Q = C_m \cdot V_m$; C_m is the membrane capacitance); therefore the state dynamics are often modeled⁴³ by $C_m \frac{dV_m}{dt} = I - g_m V_m$ (Eq. 1), where I denotes injection current and $g_m V_m$ the leaky current related to the membrane conductance g_m (Fig. 4a).

We thank the reviewer for the frank criticism to the grammatical presentation. It was our negligence not to do the most careful check before submission. We have now not only corrected the grammatical errors the reviewer pointed out, but also carefully gone through the manuscript to correct others (the efforts can be seen by looking at a Tracked version uploaded together).

We are particularly thankful to the reviewer for the helps in rewording the description of the neuron model, which is precise and easier to understand. We thus have used the reviewer's suggested version in the revised manuscript.

As this paper is primarily concerning electronic device behavior, it would probably be helpful to explain cytosolic. The paragraph below the one containing the above sentence does

explain the hypothesis “that the dynamics of filament formation in the memristor (Fig. 4b) is qualitatively analogous to the depolarization in a biological neuron”. But the explanation comes after the initial paragraph that creates the confusion.

We now have revised the manuscript to start with the description of biological process and then mechanistic analogy in memristor, before the statement of constructing artificial neuron (page 6).

Because both cyclic voltammetry (CV) and capacitance (C) are referred to in this paper, it is important that the authors always make it clear which they are referring to. I suggest that they write out the words cyclic voltammetry in the figure caption to Fig. 3.

Great point. We now have skipped the use of acronym for cyclic voltammetry to avoid the confusion.

I don't see where Figures 4A and 4B are referred to in the body of the text. What is their purpose?

We now have changed the capital “A” “B” to “a”, “b” in Fig. 4. Figures 4a and 4b were initially referred in (lines 171 and 178) and (line 177) in the main text. They were placed together to schematically show the similarity in the integrate-and-fire process between a neuron and a memristor.

Be sure that the supplemental figures have complete figure captions or accompanying notes that ensure that the presented data and its technical importance can be fully understood. Following are a few issues that were easily noticed.

SFig 4. & 5 & 7 & 8. &17. Be sure to indicate what device structure (planar or vertical) is used for these data.

4 - protein-nanowire device (nothing indicated geometry)

5 - protein-nanowire memristor (nothing indicated geometry)

6 - protein-nanowire memristors (picture indicates geometry)

7 - protein-nanowire memristor (nothing indicates geometry)

8 - protein-nanowire device (nothing indicates geometry)

9 - vertical protein-nanowire memristors (name and picture indicate geometry)

17 - protein-nanowire memristor (also protein-nanowire synapse) (nothing indicates geometry)

We now have added all the missing device details to all SI figures.

Supplementary Fig 10. Right. The vertical devices are devices are effectively two devices in parallel. What is the appropriate scaling factor? (It doesn't matter here because they are all treated similarly, but what is the appropriate length for comparing vertical to horizontal devices?)

As the reviewer pointed out, a vertical device is effectively two planar devices in parallel. Planar devices had the same width in electrode $\sim 1 \mu\text{m}$ (Fig. 3a), and the vertical devices had equivalent widths ($\times 2$) from $4 \mu\text{m}$ to $40 \mu\text{m}$ (Supplementary Fig. 10 (now SFig. 11)).

There is one difference. Vertical devices had an electrode spacing $\sim 20 \text{ nm}$ defined by the SiO_2 layer thickness, whereas planar devices had a range from 100 to 500 nm due to lithographic limit (Supplementary Fig. 7). For filamentary memristive switching, it was expected that the

switching voltage would be independent of the electrode spacing (the forming voltage would be dependent), which was consistent with the experimental observations (Supplementary Fig. 11b & Fig. 14a).

To push the limit, we have made a vertical device with an electrode size of $\sim 100 \times 100 \text{ nm}^2$ (Fig. R5a), which is equivalent to a planar device with 200 nm electrode width and 20 nm electrode spacing. So it is equivalent to a 5-fold down scaling from the typical planar devices of $1 \mu\text{m}$ width and 100 nm spacing. The device maintained the sub-100 mV switching after forming (Fig. R5b).

We have now added Fig. R5 to a new Supplementary Fig. 12 to show the further support to the filamentary mechanism and the potential in device scaling.

Fig. R5. (a) SEM image of a vertical $100 \times 100 \text{ nm}^2$ device. (b) Switching IVs (red is the forming curve).

What is the difference in the conditions for SFig. 11 and SFig13.?

A chemical ethanolamine was used during the purification of nanowires, although it was subsequently removed through dialysis. We intended to strictly exclude the possibility of effect coming from ethanolamine residue, by treating the device with normal concentration of ethanolamine (SFig. 11, now SFig. 13). SFig. 13 (now SFig. 15) was a bare device without any treatment.

SFig. 16 (b) What does the data represent? Is it the initial formation switch for five separate devices? Is it the first five cycles of a single device?

SFig. 16 (b) (now SFig. 20) is the first five cycles from a single device. The device never went down to a set voltage below 0.5 V. We have now revised the caption to avoid the confusion.

In summary, we hope our extensive efforts have substantially improved the manuscript to *Nature Communication* standard. We are truly grateful to the reviewer for the constructive comments, in which we can tell the goodwill in helping to improve the work. We also wish the reviewer a great holiday and happy new year.

Reviewers' comments:

Reviewer #1 (Remarks to the Author):

The authors have replied satisfactorily to the concerns that I raised.
I recommend publication as it is.

Reviewer #2 (Remarks to the Author):

The authors have presented a revised version of the manuscript and have extensively and thoroughly gone through all comments and addressed them. In general the work has been improved and clarified extensively. A demonstrated additional application in Figure 6 is helpful and useful for understanding the concept and future directions.

I apologise for my misunderstanding of the concept in the first round. It seems the device presented is an artificial neuron and I was under the impression that memristors are generally used as artificial synapses. Despite that the terminology depicts memristors mostly a tuneable resistor, I was not aware that "diffusive memristors" are basically resistors that change their conductance back to its original state, volatile devices based on slow kinetics (which allows for the accumulation of charges to a point it switches or "fires"). In that case, the presented "slow" and non-volatile memristor is less impressive as a building block for neuromorphic computing, as it seems other studies have done so before: Yoon et al, Nature Communications 2018, Wang et al. Nature Materials 2016, Wang et al. Nature Electronics 2018) but with relatively higher voltages. In particular these devices can be applied on specific applications only and as volatile memory elements they are not useful for multiply-accumulate operations that form the hearth of (traditional) neuromorphic hardware systems.

In light of those points, I am less convinced about the novelty and applicability of the presented device. Low voltage slow kinetics-based neuromorphic devices exist (see for instance Gkoupidenis et al. Adv Mat, 2015 and Xu et al. Science Advances 2016) and can for instance be based on organic mixed (ion/hole) conductor materials. These materials and devices seem more ideal for bio-interfacing as they are biocompatible, can be flexible, translate ionic to electrical currents and operate at low potentials. In fact, a similar concept as presented in Figure 6, has been shown with an organic synaptic transistor (see Kim et al, Science 2018 doi:10.1126/science.aao0098). The advantage of Ag-based diffusive devices based on protein nanowires is thus not fully clear to me.

More prominently presented as a leaky-integrate-and-fire artificial neuron the device is more impressive, but I wonder whether that alone warrants publication in Nature Communications, particularly when compared with already demonstrated similar diffusive devices.

Reviewer #3 (Remarks to the Author):

The revised version of this manuscript is significantly improved relative to the previous version. The authors have done a tremendous amount of work to address the initial feedback from the reviewers. In particular, the improved figure/discussion about electrochemistry and the additional experimental controls (presented in the supplemental information) strengthen the technical details of the work, and the presentation of a bio-interfacial application (heart rate monitoring in Figure 6 and in the body of the paper) improves the overall impact of the work.

I feel that all the changes made by the authors have made this paper acceptable.

Following are a couple items to consider before publication:

The authors do a nice job of addressing the main concern of Reviewer #2 about the inherent stability of these low-voltage in their response to the reviewer. I do not see changes to the manuscript. The authors should consider adding a brief discussion to the manuscript so that readers don't have the same question in their minds.

While the wording and grammar are dramatically improved, the paper could still be improved by further editorial inspection.

Response to Reviewers

To make it clear, we have used *italic* fonts for the reviewers' comments, **black** fonts for our replies, and **blue** fonts for revisions.

Reviewer #1 (Remarks to the Author):

The authors have replied satisfactorily to the concerns that I raised. I recommend publication as it is.

We greatly thank the reviewer for confirming the readily publishable quality of the work. We greatly appreciate the time, effort and help the reviewer devoted through the entire review process.

Reviewer #2 (Remarks to the Author):

The authors have presented a revised version of the manuscript and have extensively and thoroughly gone through all comments and addressed them. In general, the work has been improved and clarified extensively. A demonstrated additional application in Figure 6 is helpful and useful for understanding the concept and future directions.

We greatly thank the reviewer for confirming the fully addressing to previous comments and the substantial improvement in the work.

I apologise for my misunderstanding of the concept in the first round. It seems the device presented is an artificial neuron and I was under the impression that memristors are generally used as artificial synapses. Despite that the terminology depicts memristors mostly a tuneable resistor, I was not aware that “diffusive memristors” are basically resistors that change their conductance back to its original state, volatile devices based on slow kinetics (which allows for the accumulation of charges to a point it switches or “fires”). In that case, the presented “slow” and non-volatile memristor is less impressive as a building block for neuromorphic computing, as it seems other studies have done so before: Yoon et al, Nature Communications 2018, Wang et al. Nature Materials 2016, Wang et al. Nature Electronics 2018) but with relatively higher voltages. In particular these devices can be applied on specific applications only and as volatile memory elements they are not useful for multiply-accumulate operations that form the hearth of (traditional) neuromorphic hardware systems.

In light of those points, I am less convinced about the novelty and applicability of the presented device. Low voltage slow kinetics-based neuromorphic devices exist (see for instance Gkoupidenis et al. Adv Mat, 2015 and Xu et al. Science Advances 2016) and can for instance be based on organic mixed (ion/hole) conductor materials. These materials and devices seem more ideal for bio-interfacing as they are biocompatible, can be flexible, translate ionic to electrical currents and operate at low potentials. In fact, a similar concept as presented in Figure 6, has been shown with an organic synaptic transistor (see Kim et al, Science 2018 doi:10.1126/science.aao0098). The advantage of Ag-based diffusive devices based on protein nanowires is thus not fully clear to me.

More prominently presented as a leaky-integrate-and-fire artificial neuron the device is more impressive, but I wonder whether that alone warrants publication in Nature Communications, particularly when compared with already demonstrated similar diffusive devices.

We thank the reviewer for bringing about the concern in the limitation of diffusive memristors. This can be viewed in a *vice-versa* or complementary perspective (vs. nonvolatile memristors).

Nonvolatile memristors retain accumulative changes, mimicking the synaptic weight change that is one of the keys to implementing neuromorphic computing. However, a bio-realistic system (e.g., spiking neural network) will contain both synaptic components (e.g., weighted interconnects) and ‘somatic’ components (e.g., pulse-signal integrators). The nonvolatility that is well suited for constructing synaptic components, nonetheless, becomes disadvantageous in realizing ‘somatic’ elements that require an ‘auto-reset’ right after signal integration (e.g., repolarization after firing). Diffusive memristor provides this complementary functionality for constructing the ‘somatic’ components (e.g., artificial neuron—although we and others have also demonstrated dynamic synapses as well).

Above comparison draws analogy from static (nonvolatile) and dynamic (volatile) memory in a conventional computing architecture, in which one's weakness is complemented by the other's strength. And we do see a trend in the complementary integration of volatile and nonvolatile memristors to extend neuromorphic functions, including implementing synaptic STDP (*Nat. Mater.* 16, 101 (2017)) and learning in capacitive neural network (*Nat. Commun.* 9, 3208 (2018)), which are intrinsically difficult to implement if only using either type of memristor.

On recognizing the general importance of diffusive memristors, we summarize the importance of our work:

- (1) A bio-voltage (*e.g.*, <100 mV) memristor is qualitatively different from many low-voltage (*e.g.*, 0.5-1.0 V) memristors, including the conventional ones and transistor-based ones the reviewer cited. (Note that synaptic weight modulation only happened with $V_g > 1$ V (Fig. 3) in *Xu et al. Science Advances 2016*; the instant current (I_{sd}) response induced by ultralow V_g (Fig. 4) is a typical gate effect but not synaptic modulation). Our device provides the strictly matched signal window to real biological systems (*e.g.*, action potential of 50-120 mV). We believe that this qualitative difference will bring in a concept change that neuromorphic emulation can be beyond the functional level and toward bio-parameter matching.
- (2) The impact of the work can still go beyond the performance level. The work introduces a new concept to 'catalyze' ultralow voltage in memristors. Based on the concept and revealed mechanism, we believe that bio-voltage nonvolatile memristors can also be realized (we are working on that). Moreover, we anticipate that the work will point to generic strategies in tuning the function voltages in various memristors through additive material engineering. These efforts can lead to new frontiers of bio-voltage electronics or neuromorphic-bio interfaces.
- (3) Biocompatibility was not the initial focus of this research. Nevertheless (thanks to the reviewer's reminder), the protein composition, intrinsic flexibility and renewable production in the protein nanowires in fact could also lead to future 'green' electronics.

With that being said, we are humbled to see the fast progress in memristor/neuromorphic device fields, and by no means our device can attain many merits shown in organic memristors. Our hope is to contribute to both the breadth in memristor functions and the depth in mechanistic understanding. We have incorporated above thoughts into an added summary paragraph in our revised manuscript (adding the references the reviewer cited):

“Future bio-realistic systems (*e.g.*, spiking neural network) are expected to integrate both synaptic (*e.g.*, weighted interconnects) and somatic components (*e.g.*, pulse-signal processing units). Previous nonvolatile memristors have been demonstrated to be an ideal candidate to realize synaptic functions.¹¹ Diffusive memristors, due to the inherent mimicry to an 'auto-reset' in biological signal processing (*e.g.*, repolarization), can be advantageous in constructing the somatic pulse-processing components (*e.g.*, artificial neurons). We anticipate that the complementary integration of the two types can lead to expanded neuromorphic functions, as were initially demonstrated in synaptic functions³ and capacitive networks.¹⁵ Moreover, the concept of introducing catalytic elements in memristors and the mechanism revealed can be employed to engineer bio-voltage nonvolatile memristors, hence possibly forming the

complementary pair of volatile and nonvolatile bio-voltage memristors for constructing bio-voltage neuromorphic systems. The catalytic concept may lead to broad efforts and generic strategies in modulating/reducing function voltages in various memristors.

The protein nanowires are stable under harsh chemical and temperature conditions, providing broad options for the incorporation in electronic devices.^{27,54} Similar to organic memristors,⁵⁵⁻⁵⁷ the protein composition, intrinsic flexibility and renewable production in the protein nanowires can also provide them the biocompatibility for biointerfacing. This biocompatibility, combined with the ability of the memristors to perform bio-parameter matched computing, is expected to facilitate directly communication between electronic and biological interfaces. For example, bio-voltage memristors and neuromorphic components may be integrated in flexible substrates⁵⁷⁻⁵⁹ for tissue interfaces, enabling on-site signal processing for close-loop bioelectronics.”

Reviewer #3 (Remarks to the Author):

The revised version of this manuscript is significantly improved relative to the previous version. The authors have done a tremendous amount of work to address the initial feedback from the reviewers. In particular, the improved figure/discussion about electrochemistry and the additional experimental controls (presented in the supplemental information) strengthen the technical details of the work, and the presentation of a bio-interfacial application (heart rate monitoring in Figure 6 and in the body of the paper) improves the overall impact of the work. I feel that all the changes made by the authors have made this paper acceptable.

We greatly thank the reviewer for confirming our efforts and the substantial improvement in the work to publishable quality. As stated before, we see and appreciate the reviewer's goodwill to help us to improve the work. We greatly thank the reviewer for the time and effort devoted throughout the review process.

Following are a couple items to consider before publication:

The authors do a nice job of addressing the main concern of Reviewer #2 about the inherent stability of these low-voltage in their response to the reviewer. I do not see changes to the manuscript. The authors should consider adding a brief discussion to the manuscript so that readers don't have the same question in their minds.

We thank the reviewer for appreciating our addressing to Reviewer #2 and the suggestion to add some relevant discussion in the paper. In the revised manuscript (page 6, 2nd paragraph), we have added a succinct discussion:

“We would like to briefly discuss the functional stability in bio-voltage memristors from the mechanistic perspective. The bio-voltage memristor studied here falls into the category of electrochemical metallization cells,⁸ in which the conductance modulation requires a physical morphology change in the metallic filament. A reading voltage below the threshold switching voltage is expected to hardly perturb the conductance, because the physical evolution in the filament requires an electrochemical reduction that is largely a threshold event by overcoming the electrochemical reduction potential as discussed above.”

While the wording and grammar are dramatically improved, the paper could still be improved by further editorial inspection.

We thank the reviewer for confirming our effort in language editing. We will work closely with editorial help to further improve the presentation.

Reviewers' comments:

Reviewer #2 (Remarks to the Author):

In their rebuttal the authors have outlined in more detail the application field of volatile (and non-volatile) memristors and highlighted the challenges and opportunities. I completely agree that this is a growing field and new applications are thus expected to be found for novel devices and materials systems.

From the authors' added summary on importance of their device, it is apparent that the main novelty is the low (bio-) voltage. Although I agree that in general lower voltages are definitely beneficial, particularly for low(er) power consumption, I am not convinced that they are essential for (spiking) neural networks to operate in general, as is clear from all the other cited research. Yet, the authors claim their device will go beyond the functional level towards the bio-parameter matching, even though biocompatibility appeared not to be an initial focus of the research, as stated in the rebuttal.

If parameter matching is indeed important for interfacing these neuromorphic devices with biological ones, the authors should further clarify this and demonstrate a plausible route towards those type of applications. In fact, 50 - 100 mV seems still relatively high when real biological signals are involved, especially as the interface connection between soft biological tissues and inorganic devices is not ideal and results in high impedance and significant loss of signal.

In contrast, if the main goal is lowering power consumption, I would very much like to see a comparison with the higher voltage (state of the art) devices, especially in a broader circuit that, next to somatic devices also includes other relevant components such as complementary non-volatile memory elements as mentioned by the authors. I would expect that the neuron somatic functionality will only for a small part be responsible for the total power consumption.

Right now these two intended application fields for this particular device are mentioned only briefly and without a clear and convincing justification, to this reviewer, the novelty of the device is too weak to warrant publication in a high impact journal as Nature Communications. From the cited work and detailed further explanation, I do agree that these volatile memristors serve an important purpose in the field. Particularly when implemented as neurons with somatic functionality. However, I remain less convinced that the presented work is much more than an incremental improvement of a well-documented volatile (diffusive) memristor, by lowering the switching voltage with around 100-400 mV.

Reviewer #3 (Remarks to the Author):

In my opinion, with this version the authors have successfully addressed all my concerns and those of the other reviewers as well. I recommend publication.

Response to Reviewers

To make it clear, we have used *italic* fonts for the reviewers' comments, **black** fonts for our replies, and **blue** fonts for revisions.

Reviewers' comments:

Reviewer #2 (Remarks to the Author):

In their rebuttal the authors have outlined in more detail the application field of volatile (and non-volatile) memristors and highlighted the challenges and opportunities. I completely agree that this is a growing field and new applications are thus expected to be found for novel devices and materials systems.

We thank the reviewer for confirming our previously efforts in addressing his/her comments/concerns. We see that the reviewer has raised some additional comments. Below please find our further addressing.

From the authors' added summary on importance of their device, it is apparent that the main novelty is the low (bio-) voltage. Although I agree that in general lower voltages are definitely beneficial, particularly for low(er) power consumption, I am not convinced that they are essential for (spiking) neural networks to operate in general, as is clear from all the other cited research. Yet, the authors claim their device will go beyond the functional level towards the bio-parameter matching, even though biocompatibility appeared not to be an initial focus of the research, as stated in the rebuttal.

Please find our detailed addressing below regarding the potential benefits in both bioelectronic interfaces and neural networks. We make some quick notes here.

(1) The key to power reduction in (spiking) neural network (SNN) is that it consumes minimal energy in the idle state, which also means that ideally all power consumption goes to the active spiking events (e.g., similar to biosystem). If the goal is to reduce power at architecture level, we may not need to focus on spiking amplitude initially. However, if optimal architecture is found and we intend to push the limit (e.g., achieve bio-power level computing), then spiking amplitude becomes the dominant factor in power consumption (see details later).

(2) We also note that a direct/physical memristor-tissue contact is unlike in bioelectronic interfaces. Instead, the device body shall be passivated/sealed, leaving only certain terminal(s) for interfacing. Thus, the discussion of biocompatibility from material perspective does not necessarily lead to a direct inference from practical perspective. In addition, a biologist's 'biocompatibility' can be different and stricter than an engineer's 'biocompatibility'.

If parameter matching is indeed important for interfacing these neuromorphic devices with biological ones, the authors should further clarify this and demonstrate a plausible route towards those type of applications. In fact, 50 - 100 mV seems still relatively high when real biological signals are involved, especially as the interface connection between soft biological tissues and inorganic devices is not ideal and results in high impedance and significant loss of signal.

We thank the reviewer for pointing out (current) challenges in biointerfacing. It summarizes the requirements of both (i) parameter matching in electronics and (ii) efficient signal retrieval/restoration in biointerface to fulfill the electro-bio interfacing.

We think that we shall look at it in a progressive perspective: now that we have overcome the bottleneck in (i), it becomes more promising as we now only need to tackle the remaining challenge in (ii). In other words, the progress in (i) shall not be bound by the current challenge in (ii).

Only when (ii) were completely impossible, would we question the merit in (i). But that is not the case. 3D nanoscale engineering has yielded various intracellular biosensors, including 3D nanotransistors,¹⁻⁵ nanopipettes,^{6,7} and nanopillar electrodes,^{8,9} which were demonstrated to record close-to-full amplitude of action potentials. The high interfacial impedance in these cases was compensated by a much higher sealing impedance. For example, nanopillar electrode achieved a recording amplitude up to 99 mV (*Nano Lett.* 2017, 17, 2757), which well matched the input signal in our memristors. Admittedly, these 3D biosensors have not yet been integrated in *in vivo* bioelectronic interfaces. However, these developments provide a promising perspective in future feasibility (as another whole field has been working on that).

Above analyses indicate that the potential of implementing bio-voltage memristors in bioelectronic interface is expectable. The demonstrated artificial neurons and signal integrations all used bio-realistic signals of emulated action potentials (Figs. 4 & 6), although obtaining actual bio-signal is beyond our current capability and focus.

We have incorporated above analyses in an extended discussion in the revised manuscript (page 8):

“Cellular signal (*e.g.*, action potential) has similar amplitude,¹⁶ although the retrieval of the intracellular amplitude requires an efficient cell-sensor coupling. Recent advances in 3D nanoscale bioprobes, including nanotransistors,⁵³⁻⁵⁵ nanopipettes,^{56,57} and nanopillar electrodes,^{58,59} have successfully demonstrated close-to-full-amplitude signal recording. These advances have indicated the potential in integrating intracellular bioprobes with bio-voltage memristors in new generations of bioelectronic interfaces, in which the bio-voltage artificial neuron may form direct cellular communication with biological neurons.”

References:

- [1]. *Science* **329**, 831-834 (2010).
- [2]. *Nat. Nanotechnol.* **7**, 174-179 (2012).
- [3]. *Nano Lett.* **12**, 3329-3333 (2012).
- [4]. *Nat. Nanotechnol.* **9**, 142-147 (2014).
- [5]. *Nat. Nanotechnol.* **14**, 783-790 (2019).
- [6]. *Cell Rep.* **26**, 266-278 (2019).
- [7]. *Nat. Nanotechnol.* **12**, 335-342 (2017).
- [8]. *Nat. Methods* **7**, 200-202 (2010).
- [9]. *Nano Lett.* **17**, 2757-2764 (2017).

In contrast, if the main goal is lowering power consumption, I would very much like to see a comparison with the higher voltage (state of the art) devices, especially in a broader circuit that, next to somatic devices also includes other relevant components such as complementary non-volatile memory elements as mentioned by the authors. I would expect that the neuron somatic functionality will only for a small part be responsible for the total power consumption.

The benefit of bio-voltage operation can be a ‘both’ instead of ‘either/or’ between bioelectronic interfaces (above) and low-power computing due to their intercorrelation. Here, we provide analysis and experimental support to the potential in low-power computing.

As analyzed at the beginning, the spiking events will dominate the power consumption in an ideal SNN. The average power in a spiking event is proportional to the product of operation voltage (V_{th}) and current (typically with compliance as I_{cc}). So we start with the comparison of this basic quantity of our device with other known diffusive memristors. To be objective, the broad list (Fig. R1) is obtained from a recent review (*Adv. Funct. Mater.* 2018, 28, 1704862). The comparison shows that our device features a minimal power (60 mV \times 1 nA \sim 60 pW, Fig. 2c in main paper) at least 1000-fold lower than the lowest power in listed memristors (Fig. R1).

This initial analysis is further supported by our experimental evaluation of the energy cost ($V_{in} \cdot I_{on} \cdot t_n$) in a spiking event from a constructed neuron (Fig. R2a). The lowest dynamic energy in our device is \sim 87 fJ (with compliance) and \sim 3.4 pJ (without compliance), which again are at least two orders of magnitude lower than the dynamic energy consumed in spiking events (with similar spiking width t_n) from previous demonstrated neurons^[1-5] (Fig. R2b&c).

We further implement the bio-voltage neurons in a SNN for analysis. An exemplary SNN integrating both somatic and synaptic elements was recently demonstrated (*Nat. Electron.* 2018, 1, 137). In this SNN, each neuron made from a diffusive memristor integrates weighted input from an array

Material system	I_{HRS} or R_{HRS} (V_{read})	I_{LRS} or R_{LRS}	ON/OFF	V_{th} (DC)
Pt/TiO ₂ /Ag		1 nA (I_{cc})		40 V (forming: 80 V)
Pt/Vac/Ag ₂ S/Ag		\sim 77.5 μ S		
Pt/Cu/TaO _x /Cu	\sim 4 nA (0.2 V)	10 μ A (I_{cc})	2.5×10^3	\sim 0.25 V
Pt/a-La _{0.3} Mn _{0.7} SrO ₃ /Ag	\sim 1 nA (0.25 V)	10 μ A (I_{cc})	10^4 (0.25/0.5 V)	\sim 0.4 V
Pt/TiO ₂ /Cu	\sim 100 pA (0.25 V)	1 μ A (I_{cc})	10^4 (0.25/0.5 V)	\sim 0.4 V
Pt/SiO ₂ /Ag	\sim 1 pA (5 V)	100 nA (I_{cc})	10^5 (5/10 V)	\sim 6 V (Forming: 20 V)
Pt/Ag ₂ S/Ag/Pt		100 nA (I_{cc})		\sim 0.45 V
Pt/a-C/Cu	\sim 10 μ A	\sim 50 μ A	5	\sim 0.55 V (Forming: 1.5–2 V)
Pt/TiO ₂ /Ag	\sim 1 pA (0.15 V)	10 μ A (I_{cc})	10^7 (\sim 0.15/0.3 V)	\sim 0.24/–0.5 V
Pt/Cu ₂ O/Ag-Cu ₂ O/Cu ₂ O/Pt	\sim 1 nA (0.3 V)	1 μ A (I_{cc})	10^4 (0.3/0.6 V)	\sim 0.5/–0.5 V
Cu/Cu:HfO ₂ /	\sim 1 pA (0.3 V)	10 μ A (I_{cc})	10^7 (0.3/0.6 V)	\sim 0.4 V
Pt/a-Si:H/Ag	\sim 10 pA (0.5)	10 μ A (I_{cc})	10^6 (0.5/1 V)	\sim 0.9/–0.7 V
Pt/Ag ₂ S/Ag/Ag ₂ S/Pt	\sim 300 nA (0.15 V)	10 μ A (I_{cc})	33 (0.15/0.3 V)	\sim ±0.2 V
Pt/SiO ₂ /Cu	\sim 10 pA (0.4 V)	500 μ A (I_{cc})	5×10^7 (0.4/0.8 V)	\sim 0.6 V
Pt/SiO ₂ /Ag	\sim 1 nA (0.15 V)	10 μ A (I_{cc})	10^4 (0.15/0.3 V)	\sim 0.2 V
Pt/ZrO ₂ /Ag	\sim 300 pA (0.1 V)	10 mA (I_{cc})	3.3×10^6 (0.1/0.2 V)	\sim 0.15 V
Pt/MgO:Ag/Pt, Pt/SiO ₂ N _x :Ag/Pt, Pt/HfO ₂ :Ag/Pt	\sim 300 pA, \sim 1 nA, \sim 100 fA (0.2 V)	1 μ A, 100 μ A, 1 mA	3.3×10^3 , 10^5 , 10^{10}	\sim 0.3/0.3/0.2 V
C/SiO ₂ /Ag	\sim 1 pA (1 V)	50 μ A (I_{cc})	5×10^7 (1/2 V)	\sim 2/–0.5 V
Pt/TiO ₂ /TiN/AgTe	\sim 100 pA (0.25 V)	100 μ A (I_{cc})	10^8 (0.25/0.5 V)	\sim 0.4
p ⁺ -Si/SiO ₂ /HfO ₂ /Ag	\sim 10 pA (1 V)	100 μ A (I_{cc})	10^7 (1/2 V)	\sim 1.5 V
W/Cu ₂ S/W	\sim 100 pA (0.2 V)	10 μ A (I_{cc})	10^5 (0.2/0.4 V)	\sim 0.3 V
Pd/Ag/HfO ₂ /Ag/Pd	\sim 100 fA	\sim 100 μ A	10^9	0.5 V
Pt/SiO ₂ :Ag/Ag/Pt	\sim 100 nA	\sim 100 μ A	\sim 10 ⁶	\sim 0.3 V
Pt/HfO ₂ /Ag	\sim 0.5 pA (0.2 V)	10 μ A (I_{cc})	10^9 (0.2/0.4 V)	\sim 0.3 V

Fig. R1. List of programming I_{cc} (column 3) and V_T (column 5) in memristors, from *Adv. Funct. Mater.* 2018, 28, 1704862.

Fig. R2. Dynamic power in neuron spiking. (a) Representative integrate-and-fire process in the artificial neuron by using bio-voltage pulses (100 mV, 10 μ s width (t_n), 15 μ s period). When necessary, a current compliance (I_{cc}) is set by a compliance resistor (R_c). (b) Energy consumption (E) in each spiking event at different I_{cc} in our device, compared to values (color dots) in other reported neurons.^[1-5] (c) List of measured detailed parameters vs. reported values.

of synapses made from nonvolatile memristors, which was shown to be efficient in pattern classification with unsupervised learning. We replaced the neurons made from high-voltage diffusive memristors with our bio-voltage diffusive memristors (but maintained the similar functional voltage in the synapse network). The programming energy in each synapse during training maintained the same, but would be substantially reduced if we would use bio-voltage nonvolatile synapse/memristor (the analysis is similar to that in Fig. R2).

Thus, we may only need to focus on the readout energy in the network. Without losing generality, a 2×2 synapse matrix was used (from a fabricated 10×10 synapse network) to integrate the pre-synaptic input vector and feed into the 2 bio-voltage neurons (Fig. R3a&b). The synapses were pre-trained to different weights (Fig. 3c). Spiking amplitude of 100 mV was used for post-synaptic readouts, which were subsequently integrated by the bio-voltage neurons (owing to the bio-voltage function). The resultant different neuron firings enabled signal/pattern classification (Fig. R3d). The energy cost in each neuron in the SNN is even smaller than that of an isolated one, because the synapse network also serves as a compliance resistance to reduce the current. As analyzed before, this energy cost is at least two order of magnitude lower than previous ones implemented in SNN (*Nat. Electron.* **2018**, 1, 137).

In the revised manuscript, we have added Fig. R2 and Fig. R3 into new Supplementary Fig. 22, and Fig. 23 and summarized above analyses into an extended discussion for the potential power reduction in SNN (page 8):

“In an ideal spiking neural network (e.g., with minimal energy cost in the idle state), energy consumption shall mainly go to the spiking events. As a result, reducing the spiking amplitude is expected to play the dominant role for power reduction. Our initial studies in single device (Supplementary Fig. 22) and neural network (Supplementary Fig. 23) have shown that the achieved bio-voltage amplitude can substantially reduce the energy cost in spiking events, demonstrating the potential in improving low-power neuromorphic computing.”

References:

- [1]. *J. Appl. Phys.* **124**, 152124 (2018).
- [2]. *IEEE Electron Device Lett.* **39**, 484-487 (2018).
- [3]. *IEEE Electron Device Lett.* **39** 308-311 (2017).
- [4]. *Adv. Funct. Mater.* **27**, 1604740 (2017).
- [5]. *Nat. Electron.* **1**, 137-145 (2018).

Right now these two intended application fields for this particular device are mentioned only briefly and without a clear and convincing justification, to this reviewer, the novelty of the device is too weak to warrant publication in a high impact journal as Nature Communications. From the cited work and detailed further explanation, I do agree that these volatile memristors serve an important purpose in the field. Particularly when implemented as neurons with somatic functionality. However, I remain less convinced that the presented work is much more than an incremental improvement of a well-documented volatile (diffusive) memristor, by lowering the switching voltage with around 100-400 mV.

We believe that our extensive analyses in both fields have now provided convincing details for the justification of the potential. We thank the reviewer for the time and help to improve our work.

Reviewer #3 (Remarks to the Author):

In my opinion, with this version the authors have successfully addressed all my concerns and those of the other reviewers as well. I recommend publication.

We greatly thank the reviewer for confirming our previously efforts in fully addressing his/her comments/concerns, and other reviewers' comments. We also greatly thank the reviewer for his/her time, effort, and help throughout the process to help improving our scientific presentation.

REVIEWERS' COMMENTS:

Reviewer #2 (Remarks to the Author):

The authors have addressed my comments and highlighted all relevant points in the manuscript.